# Inflammation: A Target for Treatment in Spinal Cord Injury

**DOI:** 10.3390/cells11172692

**Published:** 2022-08-29

**Authors:** Ximena Freyermuth-Trujillo, Julia J. Segura-Uribe, Hermelinda Salgado-Ceballos, Carlos E. Orozco-Barrios, Angélica Coyoy-Salgado

**Affiliations:** 1Unidad de Investigación Médica en Enfermedades Neurológicas, Hospital de Especialidades Dr. Bernardo Sepúlveda, Centro Médico Nacional Siglo XXI, Instituto Mexicano del Seguro Social, Mexico City CP 06720, Mexico; 2Posgrado en Ciencias Biológicas, Universidad Nacional Autónoma de México, Mexico City CP 04510, Mexico; 3Subdirección de Gestión de la Investigación, Hospital Infantil de México Federico Gómez, Secretaría de Salud, Mexico City CP 06720, Mexico; 4CONACyT-Unidad de Investigación Médica en Enfermedades Neurológicas, Hospital de Especialidades Dr. Bernardo Sepúlveda, Centro Médico Nacional Siglo XXI, Instituto Mexicano del Seguro Social, Mexico City CP 06720, Mexico

**Keywords:** cytokines, hormonal therapy, macrophages, microglia, natural compounds, neutrophils, pharmaceuticals, spinal cord

## Abstract

Spinal cord injury (SCI) is a significant cause of disability, and treatment alternatives that generate beneficial outcomes and have no side effects are urgently needed. SCI may be treatable if intervention is initiated promptly. Therefore, several treatment proposals are currently being evaluated. Inflammation is part of a complex physiological response to injury or harmful stimuli induced by mechanical, chemical, or immunological agents. Neuroinflammation is one of the principal secondary changes following SCI and plays a crucial role in modulating the pathological progression of acute and chronic SCI. This review describes the main inflammatory events occurring after SCI and discusses recently proposed potential treatments and therapeutic agents that regulate inflammation after insult in animal models.

## 1. Introduction

Traumatic spinal cord injury (SCI) represents a significant cause of disability [1,2]. Depending on the severity and spinal cord anatomical level of injury, patients may experience neurological deficits ranging from loss of sensation, and bowel, bladder, sexual impairment, and autonomic dysfunction to paralysis and death [3]. Approximately 2 to 3 million people worldwide have SCI, increasing by 250,000 to 500,000 annually [4]. The leading causes of SCI are traffic accidents (50%), falls and work accidents (30%), gunshot wounds or violent crimes (11%), and sports accidents (9%) [3,5]. An SCI patient’s cumulative medical expenses range from 500,000 to 2 million USD [6]. Aside from the treatment of the injury itself, secondary costs include complications from pressure ulcers, bladder/bowel dysfunction, neuropathic pain, osteoporosis, deep vein thrombosis, cystitis, respiratory problems, pneumonia, emergency readmissions, and spinal cord injury–immune deficiency syndrome (SCI–IDS) [7,8,9].

The pathophysiology of SCI is divided into two phases: the first phase refers to the instantaneous mechanical damage generated initially [5] and the second phase includes inflammation, which involves activating resident immune cells (microglia and astrocytes) and recruiting immune cells (macrophages and neutrophils). In combination, these inflammatory reactions contribute to metabolic and cellular dysfunction of neurons, glia, and other cells at the injury site and periphery. Vascular disruption and damage to the blood–spinal cord barrier (BSCB) are significant contributors to the pathophysiology and inflammation of SCI [10,11].

The inflammatory reaction is an essential host defense mechanism. However, its effects are contradictory. Although the inflammatory process eliminates invading pathogens, removes debris, and promotes wound healing, its benefits are overshadowed by an accumulation of toxic molecules produced by inflammatory cells that damage otherwise intact tissue [12]. Inflammation in SCI involves numerous cell populations, such as astrocytes, microglia, T cells, neutrophils, monocytes, as well as non-cellular mediators [13,14,15]. Although inflammation is essential in modulating the pathological progression of acute and chronic SCI, it also regulates neuronal damage and regeneration [16].

Since the extent of secondary damage largely determines the severity of injury [17], SCI might be treatable if intervened promptly, and the high mortality rates associated with SCI could be reduced [2]. Therapeutic strategies often aim to confer neuroprotection and enhance neuroregeneration by mitigating secondary injury cascades. Based on the regulation of inflammation, some of these strategies have generated beneficial effects. In this review, we first comprehensively describe the main inflammatory events during the different stages of SCI. Second, we address potential therapies recently proposed to regulate the effects of inflammation in animal models of SCI.

## 2. Main Pathological Events in SCI by Phase

The main pathological events that occur chronologically after SCI can be divided into immediate, acute, intermediate, and chronic phases (Table 1).

### 2.1. Immediate Phase

This phase begins immediately after SCI and lasts approximately two hours [18]. It is characterized by the immediate results of the injury, including traumatic axon rupture, rapid neural and glial cell death, and spinal shock [19], resulting in loss of function [5]. 

SCI directly causes necrosis and apoptosis of vascular endothelial cells, contributing to disrupting the BSCB [20]. This disruption leads to leakage of vascular material and accumulation of various inflammatory cytokines that aggravate the inflammatory reaction, edema, and secondary injury [1,20]. An SCI study demonstrated that remote ischemic preconditioning improves BSCB integrity by upregulating occludin expression [21]. Furthermore, the secretion of collagen IV and laminin is rapidly activated as a mechanism of vascular regeneration (or neovascularization) and restriction of the inflammatory reaction [20].

### 2.2. Acute Phase

The acute phase of SCI is further divided into the early acute phase and subacute phase. The early acute phase lasts 2 to 48 h after injury and is characterized by hemorrhage, increasing edema, and inflammation. The subacute phase lasts from two days to two weeks after injury. In this phase, the phagocytic response is maximal, eliminating cellular debris from the injury site and promoting axonal growth by removing myelin debris, which acts as a growth inhibitor [5]. Recruitment of blood monocytes to the injury site begins three days after SCI and lasts for seven days. Axonal growth and scarring by anti-inflammatory macrophages start after the first week of SCI [22].

### 2.3. Intermediate and Chronic Phases

The intermediate phase (2–6 weeks after SCI) is characterized by continued maturation of astroglia scarring, axonal regeneration sprouts, and the development of cysts and syrinxes. Subsequently, the chronic phase begins 6 months after SCI and lasts for life. Wallerian degeneration of injured axons is an ongoing process after SCI, and it may take years for the severed axons and their cell bodies to be removed [5]. Cytokines also play a critical role during the early stages of Wallerian degeneration by facilitating macrophage invasion to remove axon and myelin debris, an essential precondition for axonal regeneration [23]. Additionally, local and systemic inflammatory reactions during the chronic phase stimulate cavity formation and glial scarring in the medullary parenchyma and lead to neuronal and glial death [6].

## 3. Main Inflammatory Events in SCI

Inflammation is a fundamental mechanism that arises during the secondary phase of SCI. It occurs acutely within minutes in response to traumatic injury and can last for days or be present chronically for months or years [24]. Inflammation emerges as the host immune system response, whose main objective in SCI is to stop the degeneration of adjacent healthy tissue by clearing the injury site of cellular debris from necrosis cell death to prevent a toxic environment for axonal growth [25]. The main inflammatory events in secondary SCI are summarized in Figure 1. 

### 3.1. Microglia Response

Concomitant with neutrophil invasion, microglial activation occurs immediately after SCI [9,22]. In a rat model of SCI, peak activation of microglia and macrophages occurs one to three days post-SCI, whereas the same phenomenon occurs three to seven days post-SCI in mice [26] (Figure 2). Microglia cells are relevant in the pathophysiology of SCI as they undergo intense activation in response to injury and are a prominent source of inflammatory mediators [27]. After SCI, however, the microglial population at the site and in the vicinity of the injury is reduced by mechanical impact and apoptosis [28]. 

Following SCI, toll-like receptors (TLRs) on microglial cells can detect damage-associated molecular patterns (DAMPs) released from tissue damage. Subsequently, activation and morphological hypertrophy of microglial cells occur, along with functional changes [2], including transformation to a migratory mode, retraction of processes, increased soma size, upregulation of surface antigens, and production of innate proinflammatory cytokines such as TNF-α, interleukin-1 (IL-1), interleukin-2 (IL-2), interleukin-6 (IL-6), interleukin-12 (IL-12), interleukin-18 (IL-18), and chemokines [2,12]. 

In addition, since microglia are phagocytes, they function as antigen-presenting cells [12]. Microglia exist in two primary polarized states dependent on external signals: the pro-inflammatory (M1) phenotype (classically activated) and the anti-inflammatory (M2) phenotype (alternatively activated), which produces anti-inflammatory cytokines, such as interleukin-10 (IL-10) and transforming growth factor-beta (TGF-β), to help maintain homeostasis [2,12]. Microglia have a neuroprotective role by inducing astrogliosis via IGF-1. Moreover, IGF1-expressing microglia are located between the astroglial and fibrotic scars [28]. Additionally, it is important to emphasize that microenvironmental factors can influence the eventual M1 or M2 phenotype and function of microglia [29].

**Figure 2 cells-11-02692-f002:**
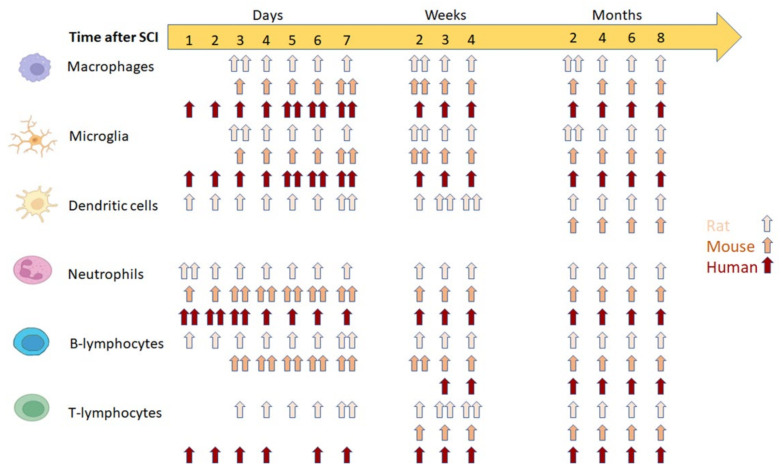
Differences in intraspinal accumulation of immune cells (macrophages, microglia, dendritic cells, neutrophils, B-lymphocytes, and T-lymphocytes) after spinal cord injury in rats, mice, and humans. Presence (one arrow) and peak (two arrows) of immune cells post-SCI in the rat (
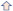
), mouse (
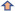
), and human (
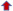
) Refs. [14,26,30,31,32,33].

In the secondary phase of SCI, activated M1 microglia initiate a cascade of neurotoxic responses and contribute to apoptosis and necrosis of endothelial cells, neurons, axons, and oligodendrocytes [2]. Activated macrophages/microglia are a chronic source of cytotoxic substances, such as TNF-α, inducible nitric oxide synthase (iNOS), hypochlorous acid (HOCl), and reactive oxygen species (ROS) [34].

With single-cell ribonucleic acid (RNA) sequencing (scRNA-seq), several microglia phenotypes with complex functional potential have been defined in both healthy and disease states [35,36,37]. In a recent study, the unique molecular signature of the cells composing the SCI site in a mouse mid-thoracic contusion model was characterized with scRNA-seq. Four microglia subtypes were identified: homeostatic microglia identified based on their expression of P2ry12, Siglech, and Tmem119, and three non-homeostatic microglia subtypes, labeled as inflammatory, dividing, and migratory microglia based on gene ontology (GO) terms for biological processes [38].

Single-cell transcriptional analyses revealed that mouse and human spinal cord microglia exist in numerous heterogeneous subpopulations, and peripheral nerve injury-induced changes in microglia were observed to differ significantly in the acute and chronic phases of neuropathic pain. In addition, sex-specific differences in gene expression were detected: a subpopulation of selectively induced microglia was identified in males but not in females three days after peripheral nerve injury. *ApoE* was also found as a highly upregulated gene in microglia in chronic phases of neuropathic pain in mice [39]. Furthermore, with transcriptional analysis of individual cells in the crush-injured spinal cord in adult mice, other authors demonstrated novel microglia and astrocyte subtypes and their dynamic conversions into additional stage-specific subtypes/states. Most of the dynamic changes occurred 3 days after injury; a recovery trend was observed between 3 and 14 days after injury, and after 14 days, reactivation of microglia and a further decrease in neuronal, astrocytic, and endothelial populations along with a continued increase in leukocyte population were evident [40].

One of the most significant inflammatory factors is ROS overproduction at the site of trauma. This ROS overproduction consumes endogenous antioxidants, disrupting the redox balance and severely damaging DNA, proteins, and lipids (lipid peroxidation). In addition to uncontrolled peroxidation, the disturbed balance contributes to inflammation and concomitant swelling [41]. 

Activated macrophages/microglia also secrete large amounts of matrix metalloproteinases (MMPs), further contributing to vascular basement membrane degradation and increased permeability for infiltrating inflammatory cells. Reactive microgliosis after SCI involves a self-perpetuating cycle initiated by neuronal damage and myelin debris activating microglia, which upregulate proinflammatory cytokines and neurotoxic molecules that cause further tissue damage, thus continuing the activation/disruption cycle [42].

### 3.2. Neutrophils

Neutrophils rapidly infiltrate the spinal cord, peak 24 h after injury, decrease to a few cells in the spinal cord in rats [43,44], and remain in the injured spinal cord for up to 6 months post-SCI [30]. Conversely, neutrophils persist in the spinal cord for several months in mice [13,26] (Figure 2). Neutrophils support recovery processes through their ability to phagocytize cellular debris and summon macrophages to the injured tissue [12]; thus, neutrophil depletion after SCI has been associated with worse outcomes [45].

Neutrophil invasion requires the presence of P-selectin, a member of the endothelial leukocyte adhesion molecule family that is rapidly expressed on the surface of endothelial cells by stimuli such as thrombin, histamine, and ROS. P-selectin is responsible for mediating early interactions between neutrophils and endothelial cells that lead to the rolling state of neutrophils on the endothelium and facilitate cell adhesion, migration, and tissue injury [46]. Thus, neutrophils gather at the site of injury and release cytokines, chemokines, and various proteases, including MMPs and neutrophil elastase, aggravating tissue edema and necrosis, and promoting apoptosis of neurons and oligodendrocytes. Consequently, these actions form a local glial scar [12,47]. 

Upon infiltrating the spinal cord, neutrophils also release ROS and nitrosyl radicals to sterilize the injured area [12,48]. These species cause uncontrolled oxidative stress. Therefore, the antioxidant defense system can no longer cope with neutralizing the free radicals produced in the organism, resulting in irreversible cellular and body damage over time [49]. Furthermore, hydrogen peroxide (H_2_O_2_) triggers the activation of neutrophils, during which the enzyme myeloperoxidase (MPO) is also secreted. Subsequently, H_2_O_2_ and MPO react with chloride (Cl^−^), creating a [H_2_O_2_--MPO-Cl^−^] system that produces HOCl. This system is paradoxically beneficial and detrimental to the host since HOCl production is necessary to kill bacteria. However, it is also one of the main factors of tissue damage following inflammatory cell activation. In addition, HOCl catalyzes the deamination and decarboxylation of amino acids [50].

Moreover, neutrophils and macrophages/microglia express the nicotinamide adenine dinucleotide phosphate (NADPH) oxidase 2 (NOX2) enzyme, a primary producer of ROS in the injured spinal cord. NOX2 transfers electrons from NADPH inside the cell, across the membrane, to extracellular oxygen to generate superoxide (O_2_^−^) [51].

Conversely, endogenous antioxidant enzymes, such as superoxide dismutase (SOD) and glutathione (GSH), contribute to the defense against ROS [52]. Superoxide (O_2_^−^) can be reduced to H_2_O_2_ by SOD and, in turn, converted to free radicals. Although SOD has been involved in neuronal recovery and regeneration, only the mitochondrial manganese SOD (MnSOD) enzyme is induced and modulated after SCI [53]. TNF-α is an upstream signal that triggers MnSOD induction through NF-kB activation. However, MnSOD can also be induced by IL-1β, IL-6, and IFN-γ [53]. 

### 3.3. Inflammasomes

Three types of inflammasomes have been characterized in microglia: the NLRP3 (NACHT, LRR, and PYD domains-containing protein 3), the NLRC4 (NLR family, CARD domain containing 4), and the AIM2 (absent in melanoma 2) inflammasomes [16]. The NLRP3 inflammasome is activated by ROS [54]. Activation of this pathway induces neuroinflammatory responses and cell death [55]. NLRPs activation leads to the recruitment of the apoptosis-associated speck-like protein (ASC) that contains a caspase activation and recruitment domain (CARD). ASC protein interacts with the CARD of pro-caspase-1, leading to its conversion from pro-caspase-1 to caspase-1 (active form). Caspase-1 converts interleukin-1 beta (IL-1β) and IL-18 pro-forms into their respective activated configuration, thus initiating an inflammatory response [56].

Treatments targeting the NLRP3 inflammasome exert neuroprotection in a rat model of contusion SCI [56]. It is essential to consider Nrf2 (nuclear factor erythroid 2-related factor 2)—a redox-sensitive master regulator [57] that modulates the transcription of several antioxidant genes under oxidative stress and inflammation [58]. In SCI, activation of Nrf2 signaling leads to detoxification and expression of antioxidant defense genes [59]. Heme oxygenase-1 (HO-1), a target of Nrf2, is a phase II detoxifying and anti-inflammatory enzyme [60] that contributes to NLRP3 inflammasome inhibition [54]. 

### 3.4. Cytokines

Proinflammatory cytokine production also increases in the immediate phase [3,61]. Although the exact mechanism of this increase in the SCI has not been explained, it is suggested that inflammatory cytokines play a central role in triggering a cascade of events leading to apoptosis [62]. Inflammatory cytokines consist of a broad category of small soluble proteins that modulate the complex function of the immune system [1]. Early accumulation of inflammatory cytokines in and around microvessels in ischemic areas can cause spinal cord edema and neuronal apoptosis [62]. 

Increased levels of IL-1β and IL-6 cause upregulation of nuclear factor kappa B (NF-κB), c-Jun N-terminal kinase (JNK), and p38 mitogen-activated protein kinase (p38MAPK), which could activate apoptosis [3]. Moreover, MAPK activation is essential for producing several inflammatory cytokines [27]. 

Additionally, IL-1β and IL-18, whose maturation in inflammatory cells depends on inflammasome activation [16], induce upregulation of IL-6 [63], thereby enhancing vascular permeability [64] and inducing neuronal apoptosis [65]. IL-1β deficiency suppresses lesion development, promotes axonal plasticity, and improves neurological outcomes [66]. In contrast, the blockade of IL-1 receptors prevents microglial activation and promotes the survival of ventral horn neurons [67] while alleviating the severity of SCI [68] and stimulating neurological recovery [69]. IL-18 is also upregulated after SCI [70], and its inhibition reduces NF-κB phosphorylation in spinal astrocytes and suppresses the expression of astrocyte markers [71]. IL-6 induces the differentiation of neural stem cells into astrocytes. However, treatment with an antibody against the IL-6 receptor suppressed injury and glial scar formation caused by inflammatory reactions after SCI in mice, improving functional recovery [72].

IL-1β and IL-6 induce iNOS in astrocytes, microglia, macrophages, and neurons. iNOS is highly involved in inflammatory processes since it produces excessive amounts of NO molecules. NO has been implicated in microglia-dependent demyelination and apoptosis of neuronal cells [2]. NO released by endothelial NOS (eNOS) is known to play a protective role. In contrast, activation of iNOS and neuronal NOS (nNOS) is followed by an overproduction of NO, which is detrimental to neuronal cells [73]. 

Anti-inflammatory cytokines, such as IL-10 and TGF-β, also play an essential role in SCI. IL-10 can be synthesized by Th2, monocytes/macrophages, astrocytes, and microglia. This cytokine suppresses the inflammatory response of monocyte/macrophages and the production of multiple cytokines, cell adhesion molecules, ROS, and nitrogen intermediates. Additionally, IL-10 affects inflammation by activating microglia/macrophages and astrocytes and reducing the production of IL-1β and iNOS. This interleukin promotes neuronal survival and functional recovery in rat and mouse models of SCI [1,74]. TGF-β is a multifunctional cytokine involved in mammal damage repair and scar tissue formation. TGF-β is a potent agent of astrocyte chemotaxis in vitro, resulting in astrocyte hypertrophy and upregulation of fibronectin and collagen IV synthesis [20].

Stromal cell-derived factor-1 (SDF-1α) is a cytokine with different biological functions, such as inflammatory cell infiltration. This cytokine is upregulated in pathological conditions [75], and its administration decreases cortical neuron death after stroke by stimulating mechanisms such as angiogenesis, neurogenesis, and modulation of the inflammatory response [76]. In addition, SDF-1α is a compound of interest for regenerative medicine because it can attract different types of stem/progenitor cells to target sites. 

Several inflammatory cytokines that participate in the pathophysiology of SCI can be beneficial or detrimental. Beneficial cytokines include erythropoietin (EPO), granulocyte colony-stimulating factor (G-CSF), granulocyte-macrophage colony-stimulating factor (GM-CSF), interferon-beta (IFN-β), interferon-gamma (IFN-γ), interleukin-4 (IL-4), IL-12, and interleukin-33 (IL-33). In contrast, cytokines that have shown detrimental behavior are chemokine C-X-C motif ligand 10 (CXCL10), interleukin-8 (IL-8), and interleukin-17 (IL-17) [1,77].

### 3.5. Interferons

TNF-α increases after SCI and may function as an apoptosis inducer in neurons and glia [78]. Different studies have shown that TNF-α can elicit trophic or toxic effects, depending on the target cell and receptors [53]. TNF-α induces and exacerbates spinal cord motoneuron death after SCI [79,80]. However, topical application of TNF-α antiserum attenuates edema, microvascular permeability, and injury in a rat model of SCI [81]. Downstream of TNF-α signaling is NF-κB, a transcriptional factor that induces inflammation-related molecules [82]. NF-κB is a redox-sensitive transcription factor that activates oxidative stress-induced-signaling cascades and plays a critical role in converting oxidative stress into inflammatory signals [83]. Within the nucleus, NF-κB induces the expression of inflammatory genes such as cyclooxygenase-2 (COX-2), iNOS, and inflammatory cytokines [84]. The NF-κB pathway also upregulates other inflammatory proteins, such as chemokines and adhesion molecules [85]. 

### 3.6. Macrophages

Macrophages, the effector cells in the inflammatory response to central nervous system (CNS) injury, are derived from microglia and hematogenous monocytes and are functionally indistinguishable in morphology and antigenic markers. In CNS injury, activated microglia are called macrophages/microglia [42]. During the first week after SCI, the populations of M1 and M2 macrophages found at the injury site are relatively evenly distributed. However, over time, macrophages eventually polarize to M1. This polarization toward M1 has been observed in different species and types of SCI, suggesting a common feature of this type of injury [86].

Peak macrophage activation is observed 7 to 14 days after injury. Macrophages function as phagocytes and serve as a reservoir for cholesterol derived from ingested myelin, which can be used to regenerate axons during remyelination [12]. 

Monocyte chemoattractant protein-1 (MCP-1) is a chemokine that regulates monocyte/macrophage migration and infiltration [87] and increases significantly after SCI [57]. IFN-γ-activated macrophages exert neurotoxic effects, whereas interleukin 4 (IL-4)-activated macrophages inhibit adverse immune responses and promote damage repair [88].

### 3.7. Dendritic Cells

Dendritic cells (DCs) are antigen-presenting cells that highly express molecular histocompatibility complex (MHC)-II proteins and proinflammatory cytokines. DCs in the spinal cord are differentiated from microglia or a circulating pool and support the ongoing inflammatory response, exacerbating the secondary injury. However, they can also produce growth factors, including neurotrophin-3, and enhance neurogenesis [12]. IL-12 is produced by DCs, macrophages, monocytes, and B cells. When secreted by DCs, IL-12 promotes functional recovery in a T8 vertebral level hemisection in female mice [89]. The recovery could be attributed to IL-12 increasing the number of activated microglia/macrophages and DCs, as well as the expression of brain-derived neurotrophic factor (BDNF), thus enhancing neurogenesis and remyelination [1].

### 3.8. Lymphocytes

B-lymphocytes are found near the injury site in the first few hours after SCI and persist for up to a week post-injury [12,24]. In contrast, peak T-lymphocyte infiltration occurs three to seven days post-SCI in rats but not in mice, in which it is delayed for up to two weeks. The population of MHC Class II-positive cells is more prevalent after SCI in mice than in rats [13,90] (Figure 2). T-lymphocytes can induce significant tissue damage through their proliferation after recognizing specific antigens, such as myelin basic protein. However, T-lymphocytes play complex roles in injury and repair mechanisms [12].

Conflicting results have been observed on the role of T- and B-lymphocytes in recovery after SCI. Whether lymphocytes become pathogenic or immunoregulatory probably depends on the nature of target antigens, antigen specificity of different lymphocyte populations, and the proximate microenvironment [91,92]. SCI impairs the development and mobilization of B-lymphocytes [93]. However, other studies demonstrate that SCI triggers activation of T- and B-lymphocytes, leading to impairments in the recovery of function and exacerbation of injury-induced pathology [94,95,96]. These effects are partly mediated by a subpopulation of autoreactive cells, some of which are myelin-reactive [97]. Furthermore, T-lymphocytes progressively increase in parallel with the activation of microglia and the influx of peripheral macrophages in the first week after injury [13].

T cells are attracted to the site of injury by the microenvironment and molecular signals. Chemokines are responsible for the migration of T cells and modulate their activation and effector potential at sites of inflammation. Depending on their phenotype (CD4+ or CD8+), T-lymphocytes can kill target cells and produce proinflammatory cytokines such as IFN-γ and TGF-β, as well as pro-regenerative trophic factors such as IL-10, IL-4, and interleukin-13 (IL-13) upon activation. Although chronic T-cell activation is precipitated during pathological fibrosis and scarring, some studies support the neuroprotective role of T-lymphocytes in models of CNS injury and neurodegeneration [98,99].

## 4. Potential Therapies to Regulate the Inflammatory Response in SCI

Although the inflammatory process generally follows a similar course in different organisms, variations have been found according to the animal model studied and to what has been observed in humans [31,32,33] (Figure 2). Knowing these differences is of utmost importance for correctly applying the proposed treatments and interpreting the outcomes and findings. With this in mind, the therapies offered to date to regulate the inflammatory response in SCI in different animal models are presented below, as well as the outcomes, advantages, and disadvantages of each treatment [100,101,102,103,104,105,106,107,108,109,110,111,112,113,114,115,116,117,118,119,120,121,122,123,124,125,126,127,128,129,130,131,132,133,134,135,136,137,138,139,140,141,142,143,144,145,146,147,148,149,150,151,152,153].

For several years, it has been known that many potential treatments and therapeutic agents for SCI regulate inflammatory cytokines and other inflammatory factors (Table 2). This review describes the results of the leading therapies tested in pre-clinical studies that appear promising for clinical use.

### 4.1. Hormone Therapy

Hormone therapy is a therapeutic approach to modulating inflammation in SCI based on administering estrogen, progesterone, or ghrelin to improve functional outcomes [3,100,101,102,103]. 

Estrogen has shown neuroprotective effects due to its anti-inflammatory effects [3]. After SCI in rats, twice-daily estrogen administration significantly reduced TNF-α and gene expression of its downstream cytokine iNOS [100], demonstrating its inhibitory effect on inflammation after SCI. Estrogen levels may be partially responsible for the improved functional outcomes in females relative to males after SCI [3]. Moreover, some estrogen receptor binding molecules—such as tamoxifen and other estrogen receptor agonists—have shown similar or identical neuroprotective functions to estrogen after SCI [3]. 

Progesterone is another anti-inflammatory hormone since it inhibits inflammatory cytokines. In a rat model of SCI, progesterone administration significantly reduced the expression of TNF-α and iNOS, which produce inflammatory mediators and NO [101]. Furthermore, progesterone reduces axonal dieback and neuronal death in mice after SCI by downregulating inflammatory cytokines, such as iNOS, MCP-1, and IL-1β, as well as activated caspase-3 and glial fibrillary acidic protein (GFAP) [102], a selective marker of astrocytes [104]. 

Moreover, ghrelin is a 28-amino acid gastric hormone with multifunctional roles in appetite, adiposity, energy balance, gastric motility, and acid secretion. In experimental autoimmune encephalomyelitis, a representative animal model of multiple sclerosis, ghrelin significantly reduced TNF-α, IL-1β, and IL-6 in the spinal cord microglia [105]. These findings suggest that ghrelin has a possible anti-inflammatory therapeutic effect on different pathologies, including SCI, where oligodendrocyte cell death and microglial activation result from the inflammatory process [103].

Despite the beneficial effects of hormone therapy, the lack of specificity and selectivity of these hormones, their short half-life, low accessibility, and rapid metabolism should be considered [106]. Another important consideration is that the therapeutic use of estradiol is limited by its peripheral hormonal actions, which increase the risk of breast, ovary, and endometrial cancer [107].

### 4.2. Cytokines and Interferons

Since cytokines and interferons play a significant role in regulating inflammation, their use has been proposed to control inflammation initiated after SCI. Yune et al. administered TNF-α in uninjured spinal cords to study its effect on SCI and observed an increase in MnSOD expression. Their results also revealed that NF-κB was activated and translocated to the nuclei of neurons and oligodendrocytes. In contrast, the administration of neutralizing antibodies against TNF-α in injured spinal cords attenuated the increased MnSOD expression and NF-κB activation. Their findings showed that TNF-α could induce MnSOD expression mediated by NF-κB activation and after SCI in rats [53].

Yaguchi et al. investigated the regulation of inflammation in SCI with cytokines. Using a mouse model of SCI, these authors evaluated the effect of topically IL-12 administered with a gel foam sponge [89]. Gelatin sponges are biodegradable, pH-neutral, sterile sponges with a porous structure that functions as a hemostatic agent [1]. These sponges are used in SCI models as a topically applied biomaterial system. The results showed that IL-12 increased activated amoeboid-like microglia/macrophages and BDNF expression. Increased remyelination and functional recovery were also observed [89]. This study provides encouraging evidence for using IL-12 as a potential treatment for SCI.

Stromal cell-derived factor-1 (SDF1α) is a notable cytokine in regeneration [56] that exerts neuroprotective effects after SCI. Moreover, its administration improved neural recovery after SCI by contusion in rats [75]. Zendedel et al. concluded that intrathecal administration of SDF1α reduces the inflammatory response in a rat model of weight compression SCI. They observed that SCI transiently increased the expression of NLRP3 and ASC inflammasomes and TNF-α, IL-1β, and IL-18. In contrast, SDF-1α administration significantly reduced these proinflammation cytokines and caspase-1 levels. Overall, this study suggests that the administration of SDF1α cytokine could decrease spinal cord damage and improve functional recovery after SCI [56].

Similar to hormone therapy, although it has shown promising results in several studies, cytokine therapy may have undesirable effects. For example, glial TNF-α undermines spinal learning, and its inhibition has been shown to rescue and restore adaptive spinal plasticity in SCI [108]. Additionally, most antitumor clinical trials involving IL-12 treatment did not show sustained antitumor responses and were associated with toxic side effects such as fever, fatigue, hematological changes, or hyperglycemia [109]. Further research is needed to fully understand the impact of decreasing the inflammatory response either with cytokines such as SDF-1α or any other therapy after SCI since components of the early inflammatory response after the injury may increase plasticity within injured or intact CNS axons [110].

### 4.3. Endogenous Components

Antioxidant enzymes such as SOD1 and catalase are highly efficient and can neutralize harmful radicals. SOD1 protects cells from constantly generated radicals and participates in antioxidant defense. Therefore, introducing antioxidant enzymes after SCI could neutralize excess ROS and promote rapid inflammation termination, thus decreasing post-traumatic neurodegeneration and neuronal death [41]. Several approaches have been developed to increase the stability and delivery of antioxidant enzymes. For example, antioxidant enzymes were encapsulated into polyionic complexes with cationic block copolymers (nanozymes). These nanozymes protect against proteases and are stable due to electrostatic interactions between the enzyme and the cation block copolymer. Encapsulated SOD1 showed stability, prolonged blood circulation, and improved recovery in animals with neurological deficits [111].

Nukolova et al. encapsulated SOD1 into polyion complexes and examined their ability to improve motor recovery in rats after contusion SCI. SOD1 was protected by either one (polycation) or two (polycation and polyanion) layers of polyions, resulting in active and stable single- or double-coat nanozymes. As a result, novel SOD1 nanoformats were developed for ROS scavenging after SCI. The multilayered polyionic SOD1 complex decreased inflammation and edema at the injury site and demonstrated efficacy in the recovery of rodents in this SCI model. Therefore, the double-coat nanozyme represents a therapeutic alternative for SCI treatment [41].

Administration of necrostatin-1 (Nec-1) is another strategy that proposes modulation of endogenous substances after SCI. Nec-1 is a receptor-interacting protein 1 (RIP1) inhibitor that prevents necroptosis (particular programmed necrosis) [112]. Wang et al. used a rat contusion model to investigate the role of Nec-1 in SCI recovery and found that Nec-1 reduced TNF-α, IL-1β, and IL-6 levels in the spinal cord 24 h after injury. Additionally, Nec1 reduced necroptosis through the recruitment of RIP1/3-MLKL (mixed lineage kinase domain-like pseudokinase) and apoptosis through the regulation of caspase 3 and Bax/Bcl-2. They also observed effective control of ROS production with Nec-1 treatment. Functional recovery and improved ethological performance after Nec-1 treatment confirmed the positive role of Nec-1 in SCI. However, whether Nec-1 can directly inhibit these cytokines or indirectly by inhibiting necrosis and apoptosis remains unknown [113].

Connexin43 (Cx43) is one of the primary gap junction proteins in CNS astrocytes. Different studies have shown that the levels of these proteins increase after SCI and that this increase may favor local glial networks involved in tissue responses to an injury. O’Carroll et al. used a rat model of moderate contusion SCI and administered a Cx43 mimetic peptide (Peptide5) directly into the lesion via an intrathecal catheter with an osmotic mini-pump. This treatment led to a significant improvement in functional recovery. Their results also showed that Peptide5 could regulate inflammation by decreasing TNF-α and IL-1β levels [114]. In addition, Peptide5 reduced Cx43 protein and increased Cx43 phosphorylation. These authors observed increased motor neuron survival, reduced astrocytosis, and activated microglia. Their results suggest that Peptide5 administration reduces secondary tissue damage after SCI and could be a potential treatment for SCI [114]. 

Epidermal growth factor receptor (EGFR) has recently attracted attention for its ability to regulate cell activation in the context of SCI. EGFR localizes to various cells in the CNS, including neurons, astrocytes, oligodendrocytes, and microglia. This receptor is activated once it binds to one of its several ligands, including epidermal growth factor (EGF) and TNF-α. Qu et al. hypothesized that regulating EGFR signaling could influence microglial activation and neuroinflammation. They administered inhibitors of EGFR signaling activation (C225 and AG1478) in a rat model of SCI using the weight-drop technique. Their results showed that blocking EGFR binding to its ligand reduces IL-1β and TNF-α production in microglia by inhibiting the EGFR/MAPK cascade. They also observed that SCI-induced overexpression of CD11b and GFAP was attenuated by treatment with C225 and AG1478. Overall, they concluded that by modulating the inflammatory response after SCI, inhibition of EGFR signaling reduces microglia and astrocyte activation, attenuates tissue edema, and improves morphological and functional recovery [27].

Similarly, Byrnes et al. determined the effect of regulating metabotropic glutamate receptor (mGluR) signaling in the CNS on inflammation after SCI. By administering a selective mGluR5 agonist called (RS)-2-chloro-5-hydroxyphenylglycine (CHPG) in moderate SCI at T9 in rats, they demonstrated that CHPG attenuated microglia-associated inflammatory responses in a dose-dependent manner, including the expression of ED1, Iba-1, Galectin-3, NADPH oxidase components, TNF-α, and inducible nitric oxide synthase. Furthermore, they observed that CHPG administration reduced lesion size and improved neurological recovery after SCI. Based on these results, these authors proposed that the observed effects might be mediated, in part, by modulation of microglia-associated inflammation [115].

Although using endogenous components as therapy has advantages since no adverse reaction or rejection by the organism is expected, the high cost of producing these components is a significant disadvantage. In addition, they show little stability on their own, leading to the need for sophisticated delivery techniques, such as nanozyme encapsulation of SOD1 [111]. In other cases, such as with Nec-1, the method of administration is very invasive (direct injected into the segmented vertebral cavity) to deliver the component to the lesion site [113], or with Peptide5 and EGFR, which were administered at the lesion site with an osmotic pump [27,114].

### 4.4. Pharmaceuticals

Given that many treatments have been proposed to regulate inflammation after SCI, it is not surprising that a large number of drugs have been evaluated. For example, meloxicam, a drug derived from enolic acid, is a potent inhibitor of prostaglandin biosynthesis under inflammatory conditions via the inhibition of COX-2 over COX-1 [116]. It has been suggested that COX-2 inhibitors may exert neuroprotective effects by reducing prostanoid and free radical synthesis or directing arachidonic acid metabolism through alternative metabolic pathways. Using a rat model of contusion SCI, Hakan et al. observed that meloxicam treatment decreased trauma-induced MPO activity (an index of neutrophil infiltration). These effects may be due directly to its antioxidant properties or its anti-inflammatory action through COX-2 inhibition. Overall, the anti-inflammatory agent meloxicam ameliorated histological and neurological deterioration after SCI. In addition to significantly inhibiting neutrophil infiltration, meloxicam inhibited free radical generation, lipid peroxidation, and DNA damage in the injured spinal cord tissue [117]. However, several side effects of meloxicam have been reported, such as abdominal pain, anemia, edema, and an increased risk of severe gastrointestinal adverse effects, including ulceration and bleeding [118]. 

Methylene blue exerts neuroprotective effects by targeting mitochondrial toxicity despite being a dye. Lin et al. characterized the effect of methylene blue on the inflammatory response of microglia after SCI. They first observed that IL-1β and IL-18 production by lipopolysaccharide (LPS)-stimulated microglia decreased with methylene blue. They also observed that methylene blue markedly reduced the appearance of NLRP3 and NLRC4 inflammasomes in stimulated microglia. These findings suggested that inhibition of NLRP3 and NLRC4 inflammasome activation decreased the production of mature IL-1β and IL-18, as cleaved caspase-1 was also reduced in methylene blue-treated microglia after stimulation. Finally, they confirmed neuroprotection and decreased overall neuroinflammation with methylene blue administration in the SCI rat model and concluded that methylene blue inhibits the inflammatory response of microglia after SCI by alleviating NLRP3 inflammasome activation [16].

### 4.5. Natural Compounds

The potential of different components of traditional medicine plants as anti-inflammatory treatments for SCI has been explored. Many elements of traditional Chinese medicine herbs, including quercetin, ligustilide, Asiatic acid, tetrandrine, Panax notoginsenoside, and plumbagin, exert anti-inflammatory effects in SCI models [119].

Quercetin is a principal flavonoid component of several Chinese herbs in foods and vegetables. This compound exhibits antioxidant, anti-inflammatory, and anticarcinogenic effects [120]. Moreover, after compression SCI, quercetin reduced neutrophil recruitment and MPO activity in rats [121]. 

Ligustilide is extracted from *Angelica sinensis* and represents one of the main active components of this plant. This compound exerts neuroprotective effects, dilates blood vessels, inhibits vascular smooth muscle cell proliferation, and has anti-cancer, anti-inflammatory, and analgesic properties. In a rat model of transection SCI, treatment with ligustilide suppressed ROS and iNOS gene expression and reduced IL-1β and TNF-α levels [122].

Asiatic acid (AA) is extracted from the Chinese herb *Centella asiatica* and exhibits antioxidant and anti-inflammatory properties. Jiang et al. observed that treatment with AA in the female rat model of SCI reduced cytokine levels, including IL-1β, IL-18, TNF-α, and IL-6, inhibited NLRP3 inflammasome activation, and increased Nrf2 and HO-1 levels [55]. 

Tetrandrine (TET) is a bis-benzylisoquinoline alkaloid extracted from the roots of *Stephania tetrandrae* S. Moore. TET shows anti-inflammatory, antioxidant, antihypertensive, cardioprotective, antitumorigenic, antinociceptive, and antidepressant properties [82]. In an oxygen-glucose-serum deprivation/reoxygenation (OGSD/R)-induced injury in rat spinal cord astrocytes mimicking hypoxic/ischaemic conditions in vivo, pretreatment with TET decreased the accumulation of TNF-α, IL-1β, and IL-6 in conditioned medium. Therefore, TET pretreatment attenuated OGSD/R-induced oxidative stress and inflammation through the phosphoinositide-3-kinase (PI3K)/protein kinase B (Akt)/NF-κB signaling pathway [82]. 

Panax notoginsenoside (PNS) is the main active compound of *Panax notoginseng*, an important traditional Chinese medicinal herb with anti-inflammatory, anti-edema, and anti-apoptosis effects. Primarily, PNS can protect neurons in animal models of cerebral IRI. Acute spinal cord IRI causes a significant up-regulation of IL-1β, IL-10, and TNF-α in rats, suggesting a dramatic infiltration of inflammatory cells into the grey and peripheral white matter. PNS treatment has been shown to prevent increased proinflammatory cytokines and reduced leukocyte activity. As a result, the inflammation-induced secondary injury was ameliorated. Additionally, PNS treatment exerted anti-apoptotic effects [62]. 

Plumbagin is extracted from the root of *Plumbago zeylanica* L. This compound is a vitamin K3 analog potent antioxidant with antiproliferative, chemopreventive, antimetastatic activities, and anti-inflammatory and analgesic effects [83]. In a rat model of SCI, plumbagin treatment modulated nuclear expression of Nrf-2 and NF-κB and reduced SCI-induced TNF-α and IL-1β. This study indicated that plumbagin is a potent suppressor of SCI-induced inflammation [83]. 

Traditional European medicine has also explored plant extracts as potential therapeutic agents for SCI. Apigenin is one of the most common flavonoids with antioxidant and anti-inflammatory properties [123]. This compound is mainly isolated from the buds and flowers of *Hypericum perforatum* [124], also known as St. John’s Wort [86]. However, it is also found in other plants and natural sources. Other apigenin-containing products include parsley, celery, thyme, celeriac, chamomile, onions, lemon balm, and oranges [124]. Contusion SCI in rats comes with a significant increase in IL-1β, TNF-α, and intercellular adhesion molecule-1 (ICAM-1) serum levels. Apigenin significantly reduced IL-1β, TNF-α, and ICAM-1 in this model, indicating that its protective effect is likely related to an anti-inflammatory action [86].

Rosemary and sage are rich in polyphenolic compounds, such as carnosol and rosmarinic acid. Carnosol has been identified as an excellent antioxidant with anticancer and anti-inflammatory properties [125] and exhibits cytoprotection through enhancing Nrf2-related antioxidant defense mechanisms [126]. Carnosol treatment followed by SCI modulated oxidative stress and inflammation by downregulation of NF-κB and COX-2 levels and upregulation of p-Akt and Nrf-2 levels. It also showed significant cytoprotection through the downregulation of proinflammatory cytokines (TNF-α, IL-6, and IL-1β). Therefore, carnosol’s antioxidant and anti-inflammatory function could have effectively suppressed oxidative stress and inflammation induced by SCI [125]. 

Rosmarinic acid—a polyphenol of the *Lamiaceae* family—is found in lemon balm and thyme, besides rosemary and sage [57]. Rosmarinic acid is a natural antioxidant and has potent anti-inflammatory effects. In a model of SCI, rosmarinic acid treatment exerted anti-inflammatory effects through the downregulation of NF-κB and cytokine levels (IL-6, IL-1β, TNF-α, and MCP-1). Furthermore, rosmarinic acid ameliorated Nrf2 downregulation caused by SCI and provided significant cytoprotection against SCI by preventing apoptosis [57]. 

Finally, tocotrienols are isomers of vitamin E with antioxidant, antitumor, and neuroprotective functions [20]. Tocotrienol treatment inhibited serum levels of NF-κB p65 unit, TNF-α, IL-1β, and IL-6, inhibiting iNOS expression, activity, and plasma NO production in rats with SCI. Moreover, tocotrienol significantly inhibited TGF-β and improved the injured spinal cord’s functional recovery and grey matter volume in rats, probably through the TGF-β, collagen type IV, and fibronectin signaling pathways [20]. 

### 4.6. Dietary Sources

Dietary modification may also be an approach to regulating inflammation after SCI. As mentioned above, apigenin is found in foods such as onions and oranges, and its protective effects on SCI are related to its anti-inflammatory actions [124]. Furthermore, allicin is a volatile oil found in a common ingredient, *Allium sativum*, commonly known as garlic. Garlic is a species of the onion genus that has been used for thousands of years for medicinal purposes [73]. Wang and Ren observed that allicin significantly reduced NF-κB and TNF-α levels in a mice model of SCI. This study demonstrated the protective effects of allicin following SCI through increasing heat shock protein 70 (HSP70) expression and NADH levels and decreasing iNOS and ROS levels [127]. 

Taurine (2-aminoethanesulfonic acid) is ubiquitously found in various tissues, including the spinal cord. Taurine modulates several neurological activities and has neuroprotective effects [128]. Increased taurine levels appear to be involved in neuroprotection and regeneration after SCI. Therefore, taurine could be beneficial as a therapeutic agent for SCI [77]. In a mice model of SCI, taurine administration decreased IL-6, COX-2, and MPO concentrations [129]. However, taurine has a strong hydrophilic and lipophobic nature and a rapid extraction rate. Therefore, some taurine analogs, such as N-chloro-taurine (NCT), have shown more effective therapeutic properties, including downregulation of cytokines (TNF-α, IL-1β, IL-6, and IL-8) and oxidative stress markers (ROS and MPO) [77]. 

### 4.7. Antibodies

Monoclonal antibodies (mAb) could be another strategy to attenuate secondary SCI damage by regulating inflammation. For example, it has been shown that mAb against P-selectin decreases the number of neutrophils at the injury site and attenuates motor impairment [46].

Bao et al. showed that treatment with mAb against the CD11d subunit of the CD11d/CD18 leukocyte integrin reduces the infiltration of neutrophils and macrophages into the spinal cord after injury and consequently reduces oxidative damage, improving neurological functions [48]. Additionally, treatment with antiCD11d mAb reduced the formation of the lipid peroxidation markers malondialdehyde (MDA) and 4-hydroxynonenal (4-HNE) [130]. 4-HNE depletion is significant in treating acute SCI, as it significantly influences secondary cell death. 4-HNE recruits phagocytic cells to the inflamed area, where treatment with anti-CD11d mAb reduces apoptosis and total cell death. As a treatment to prevent secondary damage after SCI, blocking the infiltration of activated leukocytes is superior to methods that only scavenge free radicals. Blocking leukocyte infiltration decreases free radical formation and reduces proinflammatory effects of activated leukocytes, such as cytokine production and the release of various proteases [48]. 

Infliximab, a chimeric human immunoglobulin G1 (IgG1) with a mouse variable fragment and a high affinity for TNF-α, is a novel immunomodulatory agent in many autoimmune diseases. Guven et al. explored its use in experimental ischemic spinal cord injury. They found that infliximab reduced the damage caused by ischemia-reperfusion injury and improved biochemical, histological, and neurological outcomes 48 h after the ischemic insult [131]. Although the potential of monoclonal antibodies as SCI therapy is exciting, unfortunately, these compounds are virtually unaffordable [132].

### 4.8. Genetic Modifications

Different studies have demonstrated a direct correlation between SCI and significant genetic alterations in micro ribonucleic acids (miRNAs). These non-coding RNAs can negatively regulate the expression of multiple genes by binding to the 3′-UTR of their target mRNAs, resulting in gene silencing through mRNA degradation or translational repression [133]. Therefore, the use of miRNAs as therapeutic agents for SCI has been explored in a rat model of contusion SCI. Cao et al. studied the effect of miRNA-210 by intrathecal injection of agomir-210 with osmotic pumps. Agomir-210 significantly promoted angiogenesis, attenuated lesion size, and improved functional recovery after SCI. In addition, agomir-210 reduced the levels of pro-apoptotic proteins (Bax) and cytokines (TNFα and IL-1β) and increased the levels of anti-apoptotic (Bcl-2) and anti-inflammatory (IL-10) proteins [133].

Small interfering RNAs (siRNAs) have emerged as a promising tool to control gene expression in many pathological conditions. After SCI, several target proteins have been investigated following siRNA administration [134]. Gao and Li successfully synthesized and tested siRNA-chitosan nanoparticles capable of delivering siRNA sequences with high transfection efficiency and low cytotoxicity to macrophages. They used a mouse model of SCI by compression to introduce a novel siRNA-based gene silencing strategy and reduce iNOS and NO production by targeting proinflammatory macrophages. Administration of these nanoparticles in vivo reduced the expression of iNOS and the apoptosis biomarkers Bax and Bcl-2. Their findings provide a reasonable basis for using siRNA-mediated gene silencing to reduce secondary injury propagation after SCI [134].

In 2016, Louw et al. introduced a miRNA-124 mimic into rat microglia ex vivo and in vivo using an SCI model, employing a chitosan/siRNA polyplex-like system. These authors demonstrated that miRNA-124 transfection reduced molecular histocompatibility complex-II (MHC-II), TNF-α, and ROS production in bone marrow-derived macrophages. Louw et al. altered the inflammatory response by affecting the transcriptome of local macrophages/microglia [42]. Chitosan polymer can also limit lipid peroxidation, mitigating secondary injury after SCI. Therefore, chitosan promotes functional recovery following trauma [135]. 

Viral vectors are another novel method that can regulate SCI inflammation through genetic modification. Injection of these vectors can express specific proteins at specific sites. In gene therapy studies targeting inflammatory cytokines after SCI, lentivirus is the most commonly used virus. Local application of lentivirus regulates several inflammatory cytokines, decreasing overexpression of TNFα and IL-1β and up-regulating anti-inflammatory cytokines IL-10 and IL-13, and promoting M2 polarization. The use of this virus also promotes functional recovery and facilitates neurorepair. Other viruses evaluated for regulating inflammation in SCI include herpes simplex virus (HSV), adenovirus, and poliovirus [1].

Each viral vector has unique advantages and disadvantages. However, the main drawback for retroviral and lentiviral vectors is insertional mutagenesis at the integration site, which originates from the disruption or inappropriate activation of transcription of a nearby host gene [136].

### 4.9. Cell Transplantation

Stem cells are undifferentiated and unspecialized cells with the capacity for self-renewal. Stem cell transplantation offers the potential to regulate inflammation after SCI by releasing cytokines and chemokines and facilitating functional recovery [137]. On this basis, the transplantation of cells from various sources has been evaluated in other models of SCI with promising results. The subsequent pre-clinical studies explored the potential of these stem cells. 

Mesenchymal stem cells (MSCs) reduce the inflammatory response in rodent models of SCI. MSCs transplanted directly into the SCI lesion site reduced the expression of multiple cytokines, including TNFα, IL-1β, IL-6, IL-2, IL-4, IL-12, IFN-α, and TGF-β1, among others. Transplantation of these cells also increased the expression of IL-4, IL-13, GM-CSF, and the ciliary neurotrophic factor. However, the reduced viability of the cells in the inflamed site of the SCI is the main drawback of direct MSCs transplantation [1].Bone marrow stromal cells (BMSCs) are considered an ideal cell source for treating SCI, as they can be donated for transplantation by the injured patient. Thus, immune repulsion can be prevented. BMSCs could be effectively guided to differentiate into neurons using the inverted colloidal crystal (ICC) scaffolds [138]. Yang et al. grafted two peptides to promote neurite outgrowth and cell attachment on the pore surface of ICC scaffolds and then evaluated the use of BMSBs on these peptide-modified ICC scaffolds as therapy in a rat model of contusion SCI. They observed a remarkable neuronal survival in animals treated with BMSCs in peptide-modified ICC scaffolds than in animals treated with directly administered BMSCs. In addition, they found decreased GFAP and TNF-α staining, meaning that this construct could inhibit or regulate glial scar formation and inflammation [138].Hematopoietic stem cells of umbilical cord blood have already proven helpful in treating various hematological and neurological disorders and types of cancer. Previously, Dasari et al. demonstrated that human umbilical cord blood stem cells (hUCBs) downregulate Fas and caspases, leading to functional recovery of rat hind limbs after SCI [139]. The authors further validated their data by analyzing the expression profiles of apoptotic genes in a model of moderate contusion SCI in rats transplanted with hUCBs. Their results showed efficient downregulation of these genes by hUCBs in the injured spinal cord. Treatment with hUCBs also downregulated the induced increase in TNF-α to basal levels, indicating their potential use as regulators of inflammation. Overall, this study showed that hUCBs could be an important therapeutic agent for the treatment of SCI [140].Neural stem cells (NSCs) can self-renew and generate neurons, oligodendrocytes, and astrocytes. NSCs immunoregulation and anti-inflammation effects have been demonstrated in vitro and in vivo. Transplantation of NSCs to the site of injury in the mouse contusion SCI model reduced neutrophils and regulated macrophage activation by inhibiting M1 macrophage activation. NSCs attenuated inflammatory cytokine mRNA levels, including TNF-α, IL-1β, IL-6 and IL-12. NSCs also inhibited the activation of bone marrow-derived macrophages, lessened the release cytokines such as TNF-α and IL-1β, and improved functional recovery after SCI [141].Dental stem cells (DSCs) are mesenchymal cells originating from the cranial neural crest. Yang et al. used a complete transection SCI model and injected DSCs into both sides of the injured spinal cord. Their results showed that some transplanted cells survived, differentiated into mature neurons and oligodendrocytes, and inhibited IL-1β expression to reduce inflammatory damage [142].The transplantation of stem cells from human exfoliated deciduous teeth (SHED) after SCI has also been explored. In a contusion SCI model, transplantation of SHEDs promoted functional recovery, decreased cystic cavity and glial scarring, and increased neurofilament density near the injury site. This study also demonstrated the ability of SHEDs to regulate inflammation in the context of SCI, as their transplantation reduced levels of the proinflammatory cytokine TNF-α [137].Olfactory ensheathing cells (OECs) are somatic cells whose transplantation into the injured spinal cord of rats reduced GFAP, IL-1β, and iNOS 14 days after the injury [104]. Schwann cells promote axonal regeneration in the peripheral nervous system and represent another somatic cell type whose use could repair the CNS after SCI [143]. After SCI, the co-transplantation of OECs and Schwann cells increased IL-4 and decreased IFN-γ levels, simultaneously reducing the cystic cavity area and improving motor functions [88].

Regardless of their origin, embryonic or adult, the use of stem cells has several disadvantages. Apart from ethical objections, embryonic stem cells are difficult to isolate, present a risk of rejection and high risk of teratocarcinoma, require immunosuppressive therapy, have arrhythmogenic potential, and lack specific identification markers, unlike adult stem cells [144].

### 4.10. NOX2 Inhibitors

NOX2 components are upregulated after SCI. Therefore, several studies have found that blocking NOX2 assembly can reduce inflammation and improve recovery after SCI [145]. Gp91ds-tat is a specific NOX2 inhibitor that effectively reduces ROS release from activated microglia [146]. In a rat model of moderate contusion SCI, Cooney et al. showed that acute single central administration of the NOX2 inhibitor gp91ds-tat led to significant improvements in the quantification of recovery after injury. A substantial reduction in post-injury inflammation was observed with gp91ds-tat administration, including decreases in acute and subacute neutrophil invasion and subacute macrophage/microglial populations [146]. 

Other NOX inhibitors have been used in the past, such as apocynin, a non-specific NOX inhibitor [147]. Administration of apocynin after SCI in rats significantly reduced inflammatory and oxidative stress markers, neutrophil invasion, nitrotyrosine production, and proinflammatory cytokine expression [148]. Cooney et al. also demonstrated that apocynin reduced the expression of adhesion molecules, which may explain the reduction in neutrophil invasion observed with local spinal cord injection of gp91ds-tat [146]. 

In these preclinical studies, both NOX inhibitors, gp91ds-tat and apocynin, were administered intraperitoneally, which is considered invasive [146,148].

### 4.11. Other Strategies

Other promising therapies to modulate inflammation in SCI involve photobiomodulation (PBM) [149] and the application of cerium oxide nanoparticles (CONP) [74]. Veronez et al. used a rat model of mechanical SCI produced by an impactor in which functional performance and tactile sensitivity improved after PBM at 1000 J/cm^2^. PBM at 750 and 1000 J/cm^2^ reduced CD-68 protein expression (a marker of inflammation) and the lesion volume [149]. Moreover, CONPs substantially suppressed the expression of acute inflammatory and apoptotic regulatory molecules, including iNOS, COX2, Nrf2, caspase 3, IL-1β, IL-6, and TNF-α in rats, particularly at the early stage of SCI. The regulatory functions of CONPs in SCI are promising, mainly for locomotor functions [74]. 

Furthermore, Liu et al. worked with a rat model of spinal cord contusion. They observed that the release of cytokines (IL-1β, IL-6, and TNF-α) was attenuated by administering hydrogen-rich saline (HrS) in the acute phase of SCI. HrS also suppressed astrogliosis and decreased GFAP expression after SCI. Since IL-1α, IL-6, and TNF-α are the initial triggers of reactive astrogliosis, these results support the idea that HrS can alleviate the glial scar barrier, reduce axonal injury, and promote functional recovery [150]. 

Acupuncture has long been used to treat numerous diseases in oriental medicine, including neurological dysfunction in neurodegenerative disorders [151]. Previously, clinical studies demonstrated that manual acupuncture or electroacupuncture (EA) improves motor function in CNS injuries, including stroke and SCI. EA has also attenuated ischemia-induced cerebral infarction and apoptosis [151]. Choi et al. used a rat model of contusion SCI and applied manual acupuncture to the specific acupoints GV26 and GB34 immediately after injury. Acupuncture reduced apoptotic cell death of neurons and oligodendrocytes, improving functional recovery after SCI. Decreased p38MAPK activation and expression of cytokines (TNF-α, IL-1β, and IL-6) and inflammatory mediators (iNOS, COX-2, and MMP-9) involved in neuronal and oligodendrocyte apoptotic death after SCI were also observed. These results suggest that acupuncture may be a therapeutic tool to improve the outcome of SCI in humans [151].

Other authors showed that the early inflammatory response components after injury could enhance plasticity in CNS axons [152]. Interestingly, Torres-Espín et al. hypothesized that the reintroduction of inflammation after SCI could reopen a period of plasticity, improving the efficacy of motor training [153]. They induced mild neuroinflammation by frequent injections of LPS in adult female rats with chronic SCI combined with rehabilitative motor training, one of the most effective and reliable approaches to promote motor recovery after incomplete SCI [154,155]. Their results suggested that mild neuroinflammation can increase the efficacy of intensive motor training associated with a high density of corticospinal axons sprouting in the intermediate grey matter in rats with chronic SCI. Overall, these results suggest that inflammation-induced neuroplasticity may be directed by training to form functional connections that enhance motor recovery. In contrast, the results also showed that using LPS alone could be detrimental to motor recovery [153].

Because inflammation is such a complicated process that evolves, treatments could be concomitant, combined, and specific to each stage of inflammation and SCI damage (Figure 3).

## 5. Conclusions

Inflammation is a relevant process in the pathophysiology of SCI. Therefore, its modulation would decrease tissue and functional damage and favor nerve regeneration and plasticity for recovering functions lost due to injury. However, depending on the therapeutic strategy after SCI, the inflammatory response and the anatomical and functional effects can be affected differently. 

It is vital to develop therapeutics that regulate the inflammatory response focusing on target cytokines allowing better functional recovery after SCI. In this review, we have described the main anti-inflammatory treatments tested in different models of SCI that have been proposed as potential therapies for SCI.

With a better understanding of the inflammatory process after SCI in animal models, further studies should be conducted to search for more targeted and specific yet less invasive and costly strategies so that these strategies can subsequently be applied in clinical trials. In addition to clinical studies, the cost-benefit of the treatment, its availability in healthcare institutions, the difficulty of using the treatments, and the importance of immediate or short- to medium-term application, among other factors, must be considered. Once the effects of these therapies are known in controlled clinical trials, they can be used promptly to prevent further damage due to inflammation and improve the quality of life of SCI patients.

## Figures and Tables

**Figure 1 cells-11-02692-f001:**
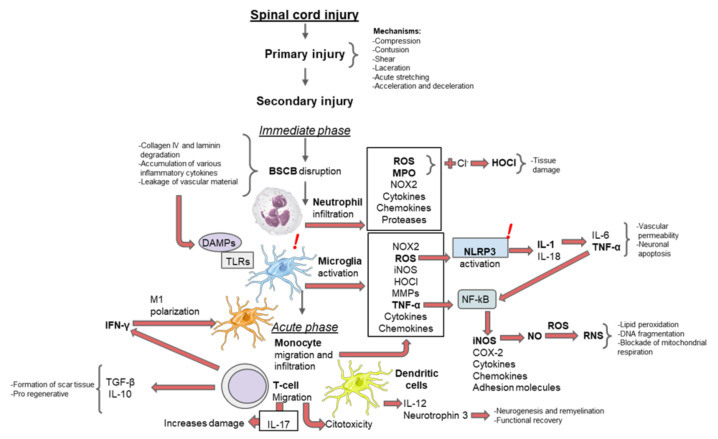
Main inflammatory events after SCI. The inflammatory mechanisms observed during secondary injury can be divided chronologically into immediate, acute, intermediate, and chronic phases. In the immediate phase, disruption of the BSCB leads to microglial activation. With microglia, neutrophils infiltrate and release proinflammatory factors that cause tissue damage, vascular permeability, neuronal apoptosis, lipid peroxidation, DNA fragmentation, and blockade of mitochondrial respiration. In the acute phase, macrophages migrate and infiltrate to promote the release of pro-inflammatory factors and help T cells to induce scar tissue formation. T cells also release pro-regenerative factors, while dendritic cells support functional recovery. Most treatments or therapeutic agents aim to regulate a particular inflammation process to improve the outcome of SCI. BSCB, blood–spinal cord barrier; Cl^−^, chloride; COX-2, cyclooxygenase-2; DAMPs, damage-associated molecular patterns; DNA, deoxyribonucleic acid; HOCl, hypochlorous acid; IFN-γ, interferon-gamma; IL-1β, interleukin-1 beta; IL-4, interleukin-4; IL-6, interleukin-6; IL-10, interleukin-10; IL-12, interleukin-12; IL-13, interleukin-13; IL-18, interleukin-18; iNOS, inducible nitric oxide synthase; MMP, metalloproteinase; MPO, myeloperoxidase; NLRP3, NACHT, LRR, and PYD domain-containing proteins 3; NF-kB, nuclear factor-kappa beta; NO, nitric oxide; NOX2, nicotinamide adenine dinucleotide phosphate (NADPH) oxidase 2; RNS, reactive nitrogen species; ROS, reactive oxygen species; TGF-β, transforming growth factor-beta; TLRs, toll-like receptors; TNF-α, tumor necrosis factor-alpha.

**Figure 3 cells-11-02692-f003:**
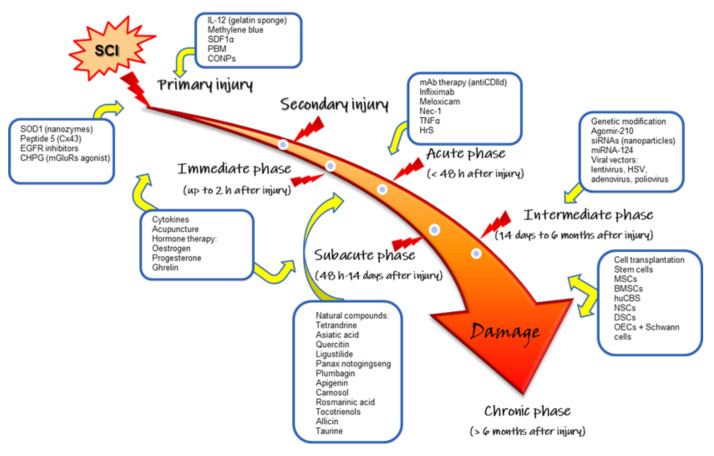
Different potential treatments were tested in animal models according to the stage or phase of spinal cord injury (SCI). After instantaneous damage (primary injury), secondary injury initiates. The immediate phase lasts two hours, followed by the acute phase up to 48 h after injury. In the sub-acute phase (14 days after injury), progressive damage occurs for up to six months (intermediate phase). After six months, when the damage is virtually irreversible, it is considered the chronic phase of SCI. BMSCs, bone marrow stromal cells; CHPG, (RS)-2-chloro-5-hydroxyphenylglycine; CONPs, cerium oxide nanoparticles; Cx43, conexin43; DSCs, dental stem cells; EGFR, epidermal growth factor receptor; HrS, hydrogen-rich saline; hUCBs, human umbilical cord blood stem cells; HSV, herpes simplex virus; mAb, monoclonal antibodies; mGluRs, metabotropic glutamate receptors; miRNAs, micro ribonucleic acids; MSCs, mesenchymal stem cells; Nec-1, necrostatin-1; NSCs, neural stem cells; OECs, olfactory ensheathing cells; PBM, photobiomodulation; SDF-1α, stromal cell-derived factor-1; siRNAs, small interfering RNAs; SOD1, superoxide dismutase 1; TNF-α, tumor necrosis factor-alpha.

**Table 1 cells-11-02692-t001:** Main pathological events in SCI by phase.

Time after SCI					
Immediate	≤2 h	≤48 h	2–13 Days	2–6 Weeks	≥6 Months
Primary Injury	Secondary Injury				
	Immediate Phase	Acute Phase	Subacute Phase	Intermediate Phase	Chronic Phase
Compression Contusion Laceration Shear Acute stretching Acceleration-deceleration Bone or disc displacement Fracture-dislocation Burst fracture Traumatic rupture of axons	Traumatic axon rupture Neural and glial cell death Spinal shock Apoptosis of vascular endothelial cells Disruption of the BSCB Accumulation of inflammatory cytokines Edema Microglia activation	ROS production Glutamate–mediated cytotoxicity Hemorrhage Edema Neutrophil invasion Early demyelination Neuronal and glial death Axonal swelling Inflammation Phagocytic response	Macrophage infiltration Initiation of astroglial scarring Blood-brain barrier repair Resolution of edema Peak phagocytic response Removal of cell debris Axonal growth Recruitment of blood monocytes to the injury site three days after SCI that lasts for seven days Axonal growth and scarring by anti-inflammatory macrophages after the first week	Lesion stabilization Continued maturation of astroglial scarring Axonal regeneration sprouts	Prolonged Wallerian Local and systemic inflammatory reactions stimulate cavity formation and glial scarring Neuronal and glial death Cytokines facilitating the invasion of macrophages to remove axon and myelin debris

ROS, reactive oxygen species.

**Table 2 cells-11-02692-t002:** Different therapies and their effects after SCI in animal models.

Therapy	Treatment	Model	Clinical Status	Pharmacological Activities	Regulation of Target Molecules	Functional and Neurological Recovery	Refs
Hormone therapy	ER agonists (estrogen, tamoxifen)	Traumatic SCI in rats	Pre-clinical	Anti-inflammatory	Reduction of TNF-α and iNOS expression	Improved BBB scores Improved MEP conductance Increase in quantity and diameter of axons	[3]
	Progesterone	SCI in rats and mice	Pre-clinical	Promyelinating, anti-inflammatory, and neuroprotective effects	Reduction of TNF-α and iNOS expression Downregulation of MCP-1, IL-1β, activated caspase-3, and GFAP	Improved motor function and histological outcomes Reduced axonal dieback and neuronal death	[101,102]
	Ghrelin	Autoimmune encephalomyelitis	Pre-clinical	Anti-inflammatory	Reduction of TNF-α, IL-1β, and IL-6 levels	Inhibited oligodendrocyte cell death Attenuated microglial activation	[103,105]
Cytokines	TNF-α	Traumatic injury in rats	Pre-clinical	Antioxidant	Increased MnSOD activity Activation and nuclear translocation of NF-κB	—	[53]
	IL-12	SCI in female mice	Pre-clinical	Functional recovery	Increased BDNF expression	Improved BBB scores Increased remyelination Induced neurogenesis	[89]
	SDF1α	SCI in male rats	Pre-clinical	Anti-inflammatory Anti-apoptosis Neuroprotection	Reduction of NLRP3, ASC, TNF-α, IL-1β, IL-18, and caspase-1 levels	Improved functional long-term recovery Attenuation of the inflammasome complex	[56]
Endogenous components	SOD1	Moderate SCI in male rats	Pre-clinical	Anti-inflammatory Antioxidant	Decreased GFAP expression Reduction of ROS	Improved locomotor functions Decreased edema	[41]
	Nec-1	Contusion SCI in rats	Pre-clinical	Anti-inflammatory, antioxidant Kinase inhibitor Anti-necroptosis and anti-apoptosis	Reduction of TNF-α, IL-1β, and IL-6 levels and ROS	Reduced ischemia lesions Induces functional recovery and neuroprotection	[113]
	Peptide 5	Moderate contusion SCI in rats	Pre-clinical	Anti-inflammatory	Reduction of TNF-α, IL-1β, and Cx43 levels Increased Cx43 phosphorylation	Improved functional recovery Increased motor neuron survival Reduced astrocytosis, activated microglia Prevented general secondary tissue damage	[114]
	EGFR inhibition by C225 and AG1478	SCI in male rats	Pre-clinical	Inhibition of the EGFR/MAPK cascade Anti-inflammatory	Decreased IL-1β, TNF-α, CD11b, and GFAP	Reduced activation of microglia and astrocytes Attenuated tissue edema Improved morphological and functional recovery	[27]
	CHPG (mGluR5 agonist)	Moderate SCI in male rats	Pre-clinical	Attenuation of microglial-associated inflammation	Reduction of ED1, Iba-1, Galectin-3, NADPH oxidase components, TNF-α, iNOS	Improved functional motor recovery Reduction of lesion size	[115]
Pharmaceuticals	Meloxicam	Contusion SCI in rats	Pre-clinical	Inhibition of COX-2 Antioxidant and anti-inflammatory	Reduction of MPO activity (neutrophil infiltration), lipid peroxidation, and DNA damage Inhibition of free radical generation	Ameliorated histological and neurological deterioration	[117]
	Methylene blue	SCI in male rats	Pre-clinical	Anti-inflammatory, anti-apoptosis, and antioxidant Targeted mitochondria toxicity Inhibition of inflammasome formation	Decreased IL-1β and IL-18, ROS, and cleaved caspase-1 levels Reduction of NLRP3 and NLRC4 activation	Ameliorated hind limb locomotor function	[16]
Natural compounds	Quercetin	Compression SCI in male rats	Pre-clinical	Antioxidant, anti-inflammatory, anti-carcinogenic Reduction of neutrophil recruitment	Reduction of MPO activity	Decreased white blood count in venous blood	[121]
	Ligustilide	Transection SCI in rats	Pre-clinical	Neuroprotection Anti-cancer, anti-inflammatory, and analgesic properties	Prevention of ROS production Suppression of iNOS expression Reduction of IL-1β, TNF-α levels	Improved BBB scale Reduced recovery of coordination Expands blood vessels Inhibition of vascular smooth muscle cells proliferation	[122]
	Asiatic acid	SCI in female rats	Pre-clinical	Antioxidant and anti-inflammatory Inhibition of NLRP3 inflammasome activation	Suppression of MPO Reduction of IL-1β, IL-18, TNF-α, and IL-6 levels Upregulation of Nrf2/HO-1 levels	Increased BBB scores Reduced inclined plane scores	[55]
	Tetrandrine	OGSD/R-induced injury in rat spinal cord astrocytes	Pre-clinical	Anti-inflammatory, antioxidant, antitumor, anti-nociceptive, and antidepressant	Decreased TNF-α, IL-1β, and IL-6 accumulation	Attenuated oxidative stress in vitro	[82]
	PNS	Acute spinal cord IRI in rats	Pre-clinical	Anti-inflammation, anti-edema, and anti-apoptosis	Prevention of IL-1β, IL-10, and TNF-α increase	Increased BBB scores Retained neurons Restored neuronal morphology Reduced leukocytes activity	[62]
	Plumbagin	SCI in male rats	Pre-clinical	Anti-proliferative, chemo-preventive, anti-metastatic, anti-inflammatory, and analgesic	Downregulation of TNF-α, IL-1β Suppression of NF-kB expression Enhancement of Nrf-2 nuclear levels	Reduced ROS and lipid peroxidation Increased antioxidant pool	[83]
	Apigenin	Traumatic SCI in rats	Pre-clinical	Antioxidant, anti-inflammatory, and anti-apoptosis	Decreased IL-1β, TNF-α, and ICAM-1 levels	Increased BBB scores Reversed changes in MDA content, SOD, and GSH-Px activity	[86]
	Carnosol	SCI in rats	Pre-clinical	Antioxidant, anticancer, and anti-inflammatory properties Enhancement of Nrf2-related antioxidant defense	Downregulation of NF-κB and COX-2 levels Decreased TNF-α, IL-6, and IL-1β levels Upregulation of p-Akt and Nrf-2 levels	Enhancement in TAC Declined TOS Reduced histological damage	[125]
	Rosmarinic acid	SCI in male rats	Pre-clinical	Antioxidant and anti-inflammatory Anti-apoptosis	Downregulation of NF-κB, IL-6, IL-1β, TNF-α, and MCP-1 levels Upregulation of Nrf-2 expression	Improved motor function Decreased oxidative stress Enhanced antioxidant status Prevented neural apoptosis	[57]
	Tocotrienol	SCI in female rats	Pre-clinical	Anti-oxidative, anti-inflammatory, anti-apoptotic, and neuroprotection functions	Reduction of NF-κB p65 unit, TNF-α, IL-1β, and IL-6 levels Decreased iNOS expression and activity and NO production Inhibition of TGF-β, collagen type IV, and fibronectin expression	Improved BBB scores Reduced volume of grey matter contusions in injured spinal cords Prevented oxidative damage	[20]
Dietary sources	Allicin	Traumatic SCI in mice	Pre-clinical	Anti-inflammatory, antibiotic, antioxidant, and antitumor	Reduction of NF-κB and TNF-α levels Inhibition of iNOS expression and ROS levels Increased CAT and SOD activity Increased HSP70 expression Elevated NADH levels	Increased BBB scores Reduced spinal cord water content	[127]
	Taurine	SCI in mice	Pre-clinical	Anti-inflammatory	Decreased IL-6 and MPO levels and COX-2 expression	Reduced neutrophil accumulation Improved functional recovery in hind limbs	[129]
Antibodies	mAb against P-selectin	Compression SCI in male rats	Pre-clinical	Anti-inflammatory	Inhibition of MPO activity P-selectin	Improved motor functions Attenuated intramedullary hemorrhages Decreased accumulation of neutrophils	[46]
	mAb against the CD11d subunit of CD11d/CD18	SCI in female rats	Pre-clinical	Reduction of neutrophil and macrophage infiltration Anti-apoptosis Anti-inflammatory and antioxidant	Decreased ED-1 and iNOS expression Reduction of MPO activity, protein nitrosylation, and lipid peroxidation Reduction of ROS, RNS, MDA, 4-HNE, and caspase-3	Improved tissue preservation and neurological function Decreased intraspinal inflammation Reduced apoptosis and cell death	[48,130]
	Infliximab	IRI to the spinal cord in male rabbits	Pre-clinical	Reduces damage caused by ischemia-reperfusion injury improves biochemical and histological outcome	Decreased MDA, GSH, and AOPP levels Increased SOD activity	Improved Tarlov scores Reduced vascular proliferation, edema, and neuronal loss	[131]
Genetic modifications	Agomir-210	Contusion SCI in male rats	Pre-clinical	Anti-inflammatory Attenuation of apoptosis	Decreased Bax, TNFα, and IL-1β Upregulation of Bcl-2 and IL-10	Promoted angiogenesis Attenuated the lesion size Improved functional recovery	[133]
	siRNA-chitosan nanoparticles	Traumatic SCI in female mice Compression SCI in guinea pigs	Pre-clinical	Anti-apoptotic, antioxidant anti-inflammatory Limitation of lipid peroxidation	Reduction of iNOS and Bax expression Increased Bcl-2 expression and NO production	Restored nerve conduction	[134,135]
	miRNA-124-chitosan polyplex	Traumatic SCI in female rats	Pre-clinical	Anti-inflammatory Limitation of lipid peroxidation	Reduction of MHC-II, TNF-α, and ROS production	Modulated macrophage/microglia activation	[42,135]
	Lentivirus	SCI in rats	Pre-clinical	Anti-inflammatory Promotion of M2 polarisation Neurorepair	Decreased TNFα and IL-1 β expression Upregulation of IL-10 and IL-13	Improved motor function	[1]
Cell transplantation	MSCs	SCI in rodents	Pre-clinical	Anti-inflammatory	Reduction of TNF-α, IL-1β, IL-6, IL-2, IL-12, IFN-α, TGF-β1, MMP-9, CCL2, CCL5, and CCL10 expression Increased IL-4, IL-13, CCL5, GM-CSF, leptin, and ciliary neurotrophic factor levels	Promoted functional recovery Limited lesion volume Less scar tissue formation	[1]
	BMSCs	SCI in male rats	Pre-clinical	Anti-inflammatory	Reduction of GFAP and TNF-α expression	Improves neuronal survival Reduced reactive gliosis	[138]
	hUCBs	SCI in male rats	Pre-clinical	Anti-apoptosis, anti-inflammatory	Downregulation of Fas expression Decreased caspases and TNF-α expression	Improved functional recovery of hind limbs Repaired spinal cord integrity	[139]
	NSCs	SCI in mice	Pre-clinical	Anti-inflammatory Immunoregulation	Attenuation of TNF-α, IL-1β, IL-6, and IL-12 mRNA levels Inhibition of iNOS expression	Improved BSA scores Reduced neutrophils Generation of neurons, oligodendrocytes, and astrocytes Inhibits the activation of M1 macrophages	[141]
	DSCs	SCI in female rats	Pre-clinical	Anti-inflammatory Neuroprotection and neuro-regeneration	Inhibition of IL-1β expression	Promoted functional recovery of hind limbs Ameliorated neural loss	[142]
	SHEDs	Contusion SCI in male rats	Pre-clinical	Anti-inflammatory	Reduction of TNF-α levels	Improved locomotor recovery Decreases cystic cavity area and glial scar formation	[137]
	OECs	Photochemical SCI in female rats	Pre-clinical	Anti-inflammatory, antioxidant	Reduction of GFAP activity, IL-1β, and iNOS levels	Reduced reactive gliosis Reduces cystic cavity area Improved neurological and electrophysiological recovery	[104]
	Co-transplantation of Schwann cells and OECs	SCI in female rats	Pre-clinical	Anti-inflammatory	Increased IL-4, IL-10, and IL-13 levels Decreased IFN-γ, IL-6 and TNF-α levels	Reduced cystic cavity area Improved motor functions	[88]
NOX inhibitors	gp91ds-tat	SCI in male rats	Pre-clinical	Anti-inflammatory, antioxidant	Inhibition of NOX2 Suppression of IL-1β, IL-6, IL-12, and TNF-α, and other pro-inflammatory cytokines Reduction of ROS	Reduced neutrophil and macrophage/microglia invasion Reduced neuronal death	[146]
	Apocynin	Compression SCI in male mice	Pre-clinical	Anti-inflammatory, antioxidant, anti-apoptotic	Blocking NADPH oxidase activation Attenuation of IL-1β, TNF-α, ICAM-1, P-selectin expression, and MPO activity Reduction of nitrotyrosine, poly-ADP-ribose, FAS ligand Prevention of Bax expression Reduction of NF-kB and P38MAPK levels Disinhibition of Bcl-2 expression Prevention of IκB-α degradation	Reduced adhesion molecule expression and neutrophil infiltration Decreased degree of tissue injury Ameliorated the loss of limb function	[148]
Other strategies	PBM	SCI in female rats	Pre-clinical	Anti-inflammatory	Decreased CD68+ cells	Improved tactile sensitivity Promoted functional recovery Reduced lesion volume	[149]
	CONPs	SCI in female rats	Pre-clinical	Anti-inflammatory, anti-apoptotic, anti-oxidative	Downregulation of iNOS, COX2, Nrf2, caspase 3, IL-1β, IL-6, and TNF-α levels Upregulation of IL-10	Reduced cavity size Improved locomotor functions	[74]
	HrS	Contusion SCI in male rats	Pre-clinical	Anti-inflammatory	Attenuation of IL-1β, IL-6, and TNF-α release Decreased GFAP, m STAT3, and p-STAT3 expression Decreased ROS production	Suppressed reactive gliosis Alleviated the glial scar barrier Reduced axonal injury Improved locomotor function	[150]
	Acupuncture	Contusion SCI in male rats	Pre-clinical	Anti-inflammatory and anti-apoptotic	Attenuation of p38MAPK activation Inhibition of caspase-3 activation Reduction of TNF-α, IL-1β, IL-6, iNOS, COX-2, and MMP-9 expression	Improved functional recovery Reduced size of the lesion cavity Attenuates ischemia-induced cerebral infarction Enhances plasticity in CNS axons	[151]
	LPS + motor rehabilitation (high-intensity training)	Chronic SCI in female rats	Pre-clinical	Inflammatory Reopening of a plasticity period	—	Increases corticospinal axons sprouting into intermediate grey matter Enhanced motor recovery in forelimbs Regain of function	[153]

AOPP, advanced oxidation protein product; BBB, Basso, Beattie, Bresnahan locomotor rating scale; BMS, Basso Mouse Scale; BMSCs, bone marrow stromal cells; CAT, catalase; CHPG, (RS)-2-chloro-5-hydroxyphenylglycine; CONPs, cerium oxide nanoparticles; Cx43, conexin43; DSCs, dental stem cells; ED-1, EGFR, epidermal growth factor receptor; GFAP, glial fibrillary acidic protein; GSH-Px, glutathione peroxidase; 4-HNE, 4-hydroxynonenal; HrS, hydrogen-rich saline; HSV, herpes simplex virus; hUCBs, human umbilical cord blood stem cells; ICAM-1, intercellular adhesion molecule-1; IL, interleukin; iNOS, inducible nitric oxide synthase; IRI, ischemia-reperfusion injury; LPS, lipopolysaccharides; mAb, monoclonal antibodies; MCP-1, monocyte chemoattractant protein-1; MDA, malondialdehyde; MEP, motor evoked potential monitoring; mGluR, metabotropic glutamate receptor; MHC-II, molecular histocompatibility complex-II; miRNAs, micro ribonucleic acids; MPO, myeloperoxidase; MSCs, mesenchymal stem cells; Nec-1, necrostatin-1; NF-κB, nuclear factor-kappa beta; Nrf-2, nuclear factor erythroid 2-related factor 2; NO, nitric oxide; NSCs, neural stem cells; OECs, olfactory ensheathing cells; OGSD/R, oxygen-glucose-serum deprivation/reoxygenation; PBM, photo-biomodulation; PNS, Panax notoginsenoside; RNS, reactive nitrogen species; ROS, reactive oxygen species; SCI, spinal cord injury; SDF1α, stromal cell-derived factor-1; siRNAs, small interfering RNAs; SOD, superoxide dismutase; TAC, total antioxidant capacity; TGF-β, transforming growth factor-β; TNF-α, tumor necrosis factor-α; TOS, total oxidant status.

## Data Availability

Not applicable.

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
