# Peer review of "Inflammation: A Target for Treatment in Spinal Cord Injury"

_cells, 2022, doi:10.3390/cells11172692_

Round 1

Reviewer 1 Report

The article from Freyermuth-Trujillo et al. is a complete and well-organized review of the status of inflammation in spinal cord research. This includes from pathophysiology to the therapeutic modulation of inflammatory markers. Certainly it represents a contribution to the field.

Author Response

  1. The article from Freyermuth-Trujillo et al. is a complete and well-organized review of the status of inflammation in spinal cord research. This includes from pathophysiology to the therapeutic modulation of inflammatory markers. Certainly it represents a contribution to the field.

Response: We greatly appreciate the time the reviewer spent reading and evaluating our review, and we are very grateful for the referee’s thorough and thoughtful comments on our paper.

Reviewer 2 Report

Review of manuscript titled “Inflammation: a target for treatment in spinal cord injury”.

Overall the review is good, but not impactful. Following are points to be considered.

Comments:

1.       Please mention whether the included studies were pre-clinical or clinical (throughout the manuscript).

2.       Give more details on the pros/cons or benefits/limitations of the different treatments.

3.       Include the clinical trial status of different treatments.

4.       Please provide a paragraph for open questions/future directions.

5.       Paragraphs are not connected well (check throughout the manuscript).

6.       Some of the paragraphs are of 2-3 lines. Please merge those paragraphs if they don’t need to be independent (specially the section 2.2.3. Lymphocytes paragraphs)

7.       References are missing lot of places (please check throughout the manuscript).

For example, line# 279-278 “B-lymphocytes are present………….to a week post-injury (ref ?).

Line# 282-284 ‘Furthermore, there are low numbers of T-lymphocytes……...first week after injury (ref ?).

8.       Please update the review with latest published articles. There is no reference from the year 2021-2022.

Author Response

Review of manuscript titled “Inflammation: a target for treatment in spinal cord injury”.

Overall the review is good, but not impactful. Following are points to be considered.

Response: We greatly appreciate the time the reviewer spent reading and evaluating our review, and we are very grateful for the referee’s thorough and thoughtful comments on our paper. We are sure that these comments helped improve the clarity of our manuscript.

Comments:

  1. Please mention whether the included studies were pre-clinical or clinical (throughout the manuscript).

Response: We thank the reviewer for this observation since it allowed us to realize there was a lack of clarity in mentioning that all the analyzed studies were conducted in animal models. Although all the studies included in the analysis were preclinical, we added a column in Table 2 mentioning the clinical status of the studies.

  1. Potential therapies to regulate the inflammatory response in SCI section, page 10, line 372.

  1. Give more details on the pros/cons or benefits/limitations of the different treatments.

Response: We thank the reviewer for this proposal. We agree that it is essential to consider the cons or limitations of the different treatments, as these are sometimes what prevent treatments from being escalated to a clinical trial stage and being administered in humans. We added a small paragraph for each treatment in the Potential therapies to regulate the inflammatory response in SCI section describing the cons and limitations as follows:

Despite the beneficial effects of hormone therapy, the lack of specificity and selectivity of these hormones, their short half-life, low accessibility, and rapid metabolism should be considered [106]. Another important consideration is that the therapeutic use of estradiol is limited by its peripheral hormonal actions, which increase the risk of breast, ovary, and endometrial cancer [107].

4.1 Hormone therapy section, page 16, line 414

Like hormone therapy, although it has shown promising results in several studies, cytokine therapy may have undesirable effects. For example, glial TNF-α undermines spinal learning, and its inhibition has been shown to rescue and restore adaptive spinal plasticity in SCI [108]. Also, most antitumor clinical trials involving IL-12 treatment did not show sustained antitumor responses and were associated with toxic side effects such as fever, fatigue, hematological changes, or hyperglycemia [109]. Further research is needed to fully understand the impact of decreasing the inflammatory response either with cytokines such as SDF-1α or any other therapy after SCI since components of the early inflammatory response after the injury may increase plasticity within injured or intact CNS axons [110].

4.2 Cytokines section, page 17, line 445

Although using endogenous components as therapy has advantages, since no adverse reaction or rejection by the organism is expected, the high cost of producing these components is a significant disadvantage. In addition, they show little stability on their own, leading to the need for sophisticated delivery techniques, such as nanozyme encapsulation of SOD1 [111]. In other cases, such as with Nec-1, the method of administration is very invasive (direct injected into the segmented vertebral cavity) to deliver the component to the lesion site [113], or with Peptide5 and EGFR, which were administered at the lesion site with an osmotic pump [27,114].

4.3 Endogenous compounds section, page 18, line 520

However, several side effects of meloxicam have been reported, such as abdominal pain, anemia, edema, and an increased risk of severe gastrointestinal adverse effects, including ulceration and bleeding [118].

4.4 Pharmaceuticals section, page 18, line 541

Although the potential of monoclonal antibodies as SCI therapy is exciting, unfortunately, these compounds are virtually unaffordable [132].

4.7 Antibodies section, page 21, line 675

Each viral vector has unique advantages and disadvantages. However, the main drawback for retroviral and lentiviral vectors is insertional mutagenesis at the integration site, which originates from the disruption or inappropriate activation of transcription of a nearby host gene [136].

4.8 Genetic modifications section, page 22, line 715

Regardless of their origin, embryonic or adult, the use of stem cells has several disadvantages. Apart from ethical objections, embryonic stem cells are difficult to isolate, present a risk of rejection and high risk of teratocarcinoma, require immunosuppressive therapy, have arrhythmogenic potential, and lack specific identification markers, unlike adult stem cells [144].

4.9 Cell transplantation section, page 23, line 778

In these preclinical studies, both NOX inhibitors, gp91ds-tat and apocynin, were administered intraperitoneally, which is considered invasive [146,148].

4.10 NOX2 inhibitors section, page 24, line 799

  1. Include the clinical trial status of different treatments.

Response: We thank the reviewer for this important observation. We agree with the reviewer that it is essential to know the clinical trial status of these treatments. However, some of them are not yet at this stage. In addition, our review described only studies in animal models, so we think this information is beyond the scope of our review. Regardless, to further clarify this point, we have added a paragraph at the beginning of the Potential therapies to regulate the inflammatory response in SCI section as follows:

Knowing these differences is of utmost importance for correctly applying the proposed treatments and interpreting the outcomes and findings. With this in mind, the therapies offered to date to regulate the inflammatory response in SCI in different animal models are presented below, as well as the outcomes, advantages, and disadvantages of each treatment.

  1. Potential therapies to regulate the inflammatory response in SCI section, page 9, line 357
  2. Please provide a paragraph for open questions/future directions.

Response: We thank the reviewer for this suggestion. We provided a paragraph on future directions as follows:

With a better understanding of the inflammatory process after SCI in animal models, further studies should be conducted to search for more targeted and specific yet less invasive and costly strategies so that these strategies can subsequently be applied in clinical trials. In addition to clinical studies, the cost-benefit of the treatment, its availability in healthcare institutions, the difficulty of using the treatments, and the importance of immediate or short- to medium-term application, among other factors, must be considered. Once the effects of these therapies are known in controlled clinical trials, they can be used promptly to prevent further damage due to inflammation and improve the quality of life of SCI patients.

  1. Conclusions section, page 25, lines 878-885.

  1. Paragraphs are not connected well (check throughout the manuscript).

Response: We apologize for this drafting error in the manuscript. We have revised the text and corrected this issue throughout the manuscript.

  1. Some of the paragraphs are of 2-3 lines. Please merge those paragraphs if they don’t need to be independent (specially the section 2.2.3. Lymphocytes paragraphs)

Response: We thank the reviewer for this observation. We have revised and restructured two main sections in the text, making the appropriate corrections and merging those short paragraphs.

  1. References are missing lot of places (please check throughout the manuscript).

For example, line# 279-278 “B-lymphocytes are present to a week post-injury (ref ?).

Response: We have checked throughout the manuscript and added the missing references as follows:

B-lymphocytes are found near the injury site in the first few hours after SCI and persist for up to a week post-injury [12,24].

3.8. Lymphocytes section, page 8, line 327

Line# 282-284 ‘Furthermore, there are low numbers of T-lymphocytes.first week after injury (ref ?).

In contrast, peak T-lymphocyte infiltration occurs three to seven days post-SCI in rats but not in mice, in which it is delayed for up to two weeks. The population of MHC Class II-positive cells is more prevalent after SCI in mice than in rats [13,87]

3.8. Lymphocytes section, page 8, line 328

  1. Please update the review with latest published articles. There is no reference from the year 2021-2022.

Response: We thank the reviewer for this observation. We have updated the manuscript with the most recent references that we found, including those from 2020 to 2022, and have highlighted them in the reference list as follows:

  1. Quadri, S.A.; Farooqui, M.; Ikram, A.; Zafar, A.; Khan, M.A.; Suriya, S.S.; Claus, C.F.; Fiani, B.; Rahman, M.; Ramachandran, A.; Armstrong, I.I.T.; Taqi, M.A.; Mortazavi, M.M. Recent update on basic mechanisms of spinal cord injury. Neurosurg Rev 2020, 43:425–441. doi: 10.1007/s10143-018-1008-3.
  2. Ding, L.; Fu, W.J.; Di, H.Y.; Zhang, X.M.; Lei, Y.T.; Chen, K.Z.; Wang, T.; Wu, H.F. Expression of long non-coding RNAs in complete transection spinal cord injury: a transcriptomic analysis. Neural Regen Res 2020, 15:1560–1567. doi: 10.4103/1673-5374.274348
  3. Lukacova, N.; Kisucka, A.; Kiss Bimbova, K.; Bacova, M.; Ileninova, M.; Kuruc, T.; Galik, J. Glial-neuronal interactions in pathogenesis and treatment of spinal cord injury. Int J Molecular Sci 2021, 22:13577. doi: 10.3390/ijms222413577
  4. Bloom, O.; Herman, P.E.; Spungen, A.M. Systemic inflammation in traumatic spinal cord injury. Exp Neurol 2020, 325:113143. doi: 10.1016/j.expneurol.2019.113143
  5. Buzoianu-Anguiano, V.; Torres-Llacsa, M.; Doncel-Pérez, E. Role of aldynoglia cells in neuroinflammatory and neuro-immune responses after spinal cord injury. Cells 2021, 10(10):2783. doi: 10.3390/cells10102783
  6. Kubick, N.; Henckell Flournoy, P.C.; Klimovich, P.; Manda, G.; Mickael, M.E. What has single-cell RNA sequencing revealed about microglial neuroimmunology? Immunity Inflamm Dis 2020, 8:825–839. doi: 10.1002/iid3.362
  7. Milich, L.M.; Choi, J.S.; Ryan, C.; Cerqueira, S.R.; Benavides, S.; Yahn, S.L.; Tsoulfas, P.; Lee, J.K. Single-cell analysis of the cellular heterogeneity and interactions in the injured mouse spinal cord. J Exp Med 2021, 218(8):e20210040. doi: 10.1084/jem.20210040

  1. Tansley, S.; Uttam, S.; Ureña Guzmán, A.; Yaqubi, M.; Pacis, A.; Parisien, A. et al. Single-cell RNA sequencing reveals time- and sex-specific responses of mouse spinal cord microglia to peripheral nerve injury and links ApoE to chronic pain. Nat Commun 2022, 13:843. doi: 10.1038/s41467-022-28473-8
  2. Li, C.; Wu, Z.; Zhou, L. et al. Temporal and spatial cellular and molecular pathological alterations with single-cell resolution in the adult spinal cord after injury. Sig Transduct Target Ther 2022, 7:65. doi: 10.1038/s41392-022-00885-4
  3. Carpenter, R.S.; Marbourg, J.M.; Brennan, F.H.; Mifflin, K.A.; Hall, J.C.E.; Jiang, R.R.; Mo, X.M.; Karunasiri, M.; Burke, M.H.; Dorrance, A.M.; Popovich, P.G. Spinal cord injury causes chronic bone marrow failure. Nat Commun 2020, 11(1):3702. doi: 10.1038/s41467-020-17564-z
  4. Zrzavy, T.; Schwaiger, C.; Wimmer, I.; Berger, T.; Bauer, J.; Butovsky, O.; Schwab, J. M.; Lassmann, H.; Höftberger, R. Acute and non-resolving inflammation associate with oxidative injury after human spinal cord injury. Brain 2021, 144(1):144–161. doi: 10.1093/brain/awaa360
  5. Chio, J.; Xu, K.J.; Popovich, P.; David, S.; Fehlings, M.G. Neuroimmunological therapies for treating spinal cord injury: evidence and future perspectives. Exp Neurol 2021, 341:113704. doi: 10.1016/j.expneurol.2021.113704
  6. Khalil, N.Y.; Aldosari, K.F. Meloxicam. Profiles of Drug Substances, Excipients, and Related Methodology 2020, 45:159–197. doi: 10.1016/bs.podrm.2019.10.006

Reviewer 3 Report

The author tried to review the inflammatory changes after the spinal cord injury. It was organized by the role of diverse immune cells in different phases after the initial damage. The author summarised the main inflammatory events in a timely order and introduced the potential therapies. The encyclopedia-style introduction, however, lacks novelty and specific focus, which is key to a successful review paper. 

In general, the manuscript could be better organized. First, the tissue pathogenic changes, involving immune cell types, cellular receptors, signaling pathways, and inflammatory mediators should not be listed in a parallel manner. For example, in chapter 2.1, the microglia and neutrophils are involved immune cells and should be discussed together. The inflammasome activation should be under the microglia/macrophage response, and the cytokine production is also the result of the overactivation of microglia/macrophage. Second, we believe it is not appropriate to limit the role of microglia and neutrophils in a time scale of hours. In the animal model, both microglia and neutrophils contributed to the pathogenesis of SCI for days but not hours (1). Their function and phenotype will transit through the progression of the disease in a much more complicated way than traditional M1 and M2 classification (2,3). The author should introduce recent papers on more sophisticated phenotypic changes in the immune cells as a review paper. We suggest the author could use immediate, acute, and chronic phases to introduce the pathogenic changes both at the tissue and cellular levels. The involvement of immune cells and their functional transition should be introduced in a more disease-specific context. For example, the neutrophil is believed to be a significant driver of secondary damage after the initial injury. However, neutrophils are also necessary for tissue repair in the SCI animal model. The divergent effects of microglia and neutrophils are largely ignored in this review. Third, the review introduced too many regular functions of each immune cell but lacked details in the SCI disease context. Fourth, the review lacks recent progress on the systematic role of the immune system in SCI. As we know, the SCI immune activation is often characterized as local overaction while systemic suppression. The local hyperactive inflammation is sterile, while the hypoactive systemic immune response is often related to infection (especially in the lung). The divergent immune functions (time and location) become an obstacle for translational research to develop effective immunomodulatory therapies. However, the author didn't give enough attention to this. Currently, the immune activation studies in SCI are beyond the local injured spinal core. The spinal cord-gut-immune axis and spinal cord-lung infection are involved in many immune dysregulations. 

In detail, some inaccurate descriptions should be improved. For example, in line 148, the author stated that NF-kB and MAPK are apoptotic agents, but their functions include proinflammatory activation, migration, differentiation, and proliferation. We wonder if it is correct to call them apoptotic agents. In line 51, the specific cytokine TNF-alpha should be listed with other specific factors but not the general category, such as interferons or interleukins. There are more than 38 interleukins and three types of interferons. Please be specific.

Similarly, in line 54, antibodies are the effector of B lymphocytes, and cytokines are produced both from microglia/macrophage and lymphocytes. The author should not list them in a parallel manner. In line 66, there will be no drug without side effects. 

(1) Li, Chen, et al. "Temporal and spatial cellular and molecular pathological alterations with single-cell resolution in the adult spinal cord after injury." Signal transduction and targeted therapy 7.1 (2022): 1-15.

(2)Tansley, Shannon, et al. "Single-cell RNA sequencing reveals time-and sex-specific responses of mouse spinal cord microglia to peripheral nerve injury and links ApoE to chronic pain." Nature communications 13.1 (2022): 1-16.

(3)Milich, Lindsay M., et al. "Single-cell analysis of the cellular heterogeneity and interactions in the injured mouse spinal cord." Journal of Experimental Medicine 218.8 (2021): e20210040.

Author Response

Reviewer #3

The author tried to review the inflammatory changes after the spinal cord injury. It was organized by the role of diverse immune cells in different phases after the initial damage. The author summarised the main inflammatory events in a timely order and introduced the potential therapies. The encyclopedia-style introduction, however, lacks novelty and specific focus, which is key to a successful review paper.

Response: We thank the reviewer’s thoughtful comments for helping us improve our review and preventing it from lacking novelty and specific focus.

Comments:

  1. In general, the manuscript could be better organized. First, the tissue pathogenic changes, involving immune cell types, cellular receptors, signaling pathways, and inflammatory mediators should not be listed in a parallel manner. For example, in chapter 2.1, the microglia and neutrophils are involved immune cells and should be discussed together. The inflammasome activation should be under the microglia/macrophage response, and the cytokine production is also the result of the overactivation of microglia/macrophage.

Response: We thank the reviewer for suggesting better organizing our manuscript. Based on this suggestion, we restructured section 2 of the previous manuscript. In this revised version, we have divided the main events in terms of the phases of SCI (section 2) and the main inflammatory events in SCI (section 3).

  1. Main pathological events in SCI by phase section, pages 2-3, lines 64-101.
  2. Main inflammatory events in SCI section, pages 4-9, lines 102-363.
  3. Second, we believe it is not appropriate to limit the role of microglia and neutrophils in a time scale of hours. In the animal model, both microglia and neutrophils contributed to the pathogenesis of SCI for days but not hours (1).

Response: Thank you for bringing these errors to our attention. We have corrected the information as follows:

Most of the dynamic changes occurred 3 days after injury; a recovery trend was observed between 3 and 14 days after injury, and after 14 days, reactivation of microglia and a further decrease in neuronal, astrocytic, and endothelial populations along with a continued increase in leukocyte population were evident [36]( Li et al., 2022)

3.1. Microglia response section, page 5, lines 173-177.

(1) Li, Chen, et al. "Temporal and spatial cellular and molecular pathological alterations with single-cell resolution in the adult spinal cord after injury." Signal transduction and targeted therapy 7.1 (2022): 1-15.

  1. Their function and phenotype will transit through the progression of the disease in a much more complicated way than traditional M1 and M2 classification (2,3). The author should introduce recent papers on more sophisticated phenotypic changes in the immune cells as a review paper.

With single-cell ribonucleic acid (RNA) sequencing (scRNA-seq), several microglia phenotypes with complex functional potential have been defined in both healthy and disease states [31-33]. In a recent study, the unique molecular signature of the cells composing the SCI site in a mouse mid-thoracic contusion model was characterized with scRNA-seq. Four microglia subtypes were identified: homeostatic microglia identified based on their expression of P2ry12, Siglech, and Tmem119, and three non-homeostatic microglia subtypes, labeled as inflammatory, dividing, and migratory microglia based on gene ontology (GO) terms for biological processes [34](Milich, et al., 2021).

Single-cell transcriptional analyses revealed that mouse and human spinal cord microglia exist in numerous heterogeneous subpopulations, and peripheral nerve injury-induced changes in microglia were observed to differ significantly in the acute and chronic phases of neuropathic pain. In addition, sex-specific differences in gene expression were detected: a subpopulation of selectively induced microglia was identified in males but not in females three days after peripheral nerve injury. ApoE was also found as a high upregulated gene in microglia in chronic phases of neuropathic pain in mice [35] (Tansley et al., 2022).

3.1. Microglia response section, page 5, lines 156-170.

(2)Tansley, Shannon, et al. "Single-cell RNA sequencing reveals time-and sex-specific responses of mouse spinal cord microglia to peripheral nerve injury and links ApoE to chronic pain." Nature communications 13.1 (2022): 1-16.

(3)Milich, Lindsay M., et al. "Single-cell analysis of the cellular heterogeneity and interactions in the injured mouse spinal cord." Journal of Experimental Medicine 218.8 (2021): e20210040..

  1. We suggest the author could use immediate, acute, and chronic phases to introduce the pathogenic changes both at the tissue and cellular levels.

Response: We thank the reviewer for this suggestion. As mentioned earlier, we divided the main events in terms of the phases of SCI (section 2) and added a table (Table 1).

  1. Main pathological events in SCI by phase

The main pathological events that occur chronologically after SCI can be divided into immediate, acute, intermediate, and chronic phases (Table 1).

  1. Main pathological events in SCI by phase section, pages 2-3, lines 64-101.
  2. The involvement of immune cells and their functional transition should be introduced in a more disease-specific context. For example, the neutrophil is believed to be a significant driver of secondary damage after the initial injury. However, neutrophils are also necessary for tissue repair in the SCI animal model. The divergent effects of microglia and neutrophils are largely ignored in this review.

Response: We thank the reviewer for this suggestion. Although we agree with the reviewer that this is a very important topic, the length of the review did not allow us to go much deeper into the divergent effects of microglia and neutrophils. However, we add brief information on this subject as follows:

Microglia have a neuroprotective role by inducing astrogliosis via IGF-1. Moreover, IGF1-expressing microglia are located between the astroglial and fibrotic scars [28]. Also, it is important to emphasize that microenvironmental factors can influence the eventual M1 or M2 phenotype and function of microglia [29].

3.1. Microglia response section, page 5, lines 148-151.

Neutrophils support recovery processes through their ability to phagocytize cellular debris and summon macrophages to the injured tissue [12]; thus, neutrophil depletion after SCI has been associated with worse outcomes [42].

3.2. Neutrophils section, page 6, lines 194-196 and the whole section.

  1. Third, the review introduced too many regular functions of each immune cell but lacked details in the SCI disease context.

Response: We thank the reviewer for this suggestion. Throughout the text, we have included information to clarify when we are referring to events or treatments in the context of SCI. Additionally, we highlighted the importance of some immune cells during the course of SCI in Figure 2.

  1. Potential therapies to regulate the inflammatory response in SCI section, page 9, line 371.

Also, we summarized the treatments that modulate inflammation and the effects on neurological and functional recovery in Table 2.

  1. Potential therapies to regulate the inflammatory response in SCI section, page 10-15.
  2. Fourth, the review lacks recent progress on the systematic role of the immune system in SCI. As we know, the SCI immune activation is often characterized as local overaction while systemic suppression. The local hyperactive inflammation is sterile, while the hypoactive systemic immune response is often related to infection (especially in the lung). The divergent immune functions (time and location) become an obstacle for translational research to develop effective immunomodulatory therapies. However, the author didn't give enough attention to this. Currently, the immune activation studies in SCI are beyond the local injured spinal cord. The spinal cord-gut-immune axis and spinal cord-lung infection are involved in many immune dysregulations.

Response: We thank the reviewer for this important observation. We added a brief paragraph mentioning the point made by the reviewer as follows:

Aside from the treatment of the injury itself, secondary costs include complications from pressure ulcers, bladder/bowel dysfunction, neuropathic pain, osteoporosis, deep vein thrombosis, cystitis, respiratory problems, pneumonia, emergency readmissions, and spinal cord injury-immune deficiency syndrome (SCI-IDS) [7-9]

  1. Introduction section, page 1, lines 34-38.
  2. In detail, some inaccurate descriptions should be improved. For example, in line 148, the author stated that NF-kB and MAPK are apoptotic agents, but their functions include proinflammatory activation, migration, differentiation, and proliferation. We wonder if it is correct to call them apoptotic agents.

Response: In the revised version of the manuscript, we corrected this inaccuracy and rephrased it as follows:

Increased levels of IL-1β and IL-6 cause upregulation of nuclear factor kappa B (NF-κB), c-Jun N-terminal kinase (JNK), and p38 mitogen-activated protein kinase (p38MAPK), which could activate apoptosis [3]. Besides, MAPK activation is essential for producing several inflammatory cytokines [27].

3.4. Cytokines section, page 7, lines 253-256.

  1. In line 51, the specific cytokine TNF-alpha should be listed with other specific factors but not the general category, such as interferons or interleukins. There are more than 38 interleukins and three types of interferons. Please be specific.

Response: According to the reviewer's suggestion, the information on TNF-alpha was removed from section 3.4 Cytokines (page 7) and added in a new section as follows:

TNF-α increases after SCI and may function as an apoptosis inducer in neurons and glia [75]. Different studies have shown that TNF-α can elicit trophic or toxic effects, depending on the target cell and receptors [50]. TNF-α induces and exacerbates spinal cord motoneuron death after SCI [76,77]. However, topical application of TNF-α antiserum attenuates edema, microvascular permeability, and injury in a rat model of SCI [78]. Downstream of TNF-α signaling is NF-κB, a transcriptional factor that induces inflammation-related molecules [79]. NF-κB is a redox-sensitive transcription factor that activates oxidative stress-induced-signaling cascades and plays a critical role in converting oxidative stress into inflammatory signals [80]. Within the nucleus, NF-κB induces the expression of inflammatory genes such as cyclooxygenase-2 (COX-2), iNOS, and inflammatory cytokines [81]. The NF-κB pathway also upregulates other inflammatory proteins, such as chemokines and adhesion molecules [82].

3.5 Interferons section, page 8, lines 300-311.

  1. Similarly, in line 54, antibodies are the effector of B lymphocytes, and cytokines are produced both from microglia/macrophage and lymphocytes. The author should not list them in a parallel manner.

Response: We thank the reviewer for this suggestion. This sentence has been removed to avoid any misunderstanding.

  1. In line 66, there will be no drug without side effects.

Response: We apologize for this mistake. We agree with the reviewer’s comment, so we have removed this statement.

Round 2

Reviewer 2 Report

The authors’ response and revisions have satisfactorily addressed my comments.